# Statistical modeling of adaptive neural networks explains co-existence of avalanches and oscillations in resting human brain

**Fabrizio Lombardi** [1]✉, **Selver Pepić**[1], **Oren Shriki** [2], **Gašper Tkačik**[1] ✉ & **Daniele De Martino** [3]✉

Neurons in the brain are wired into adaptive networks that exhibit collective dynamics as diverse as scale-specific oscillations and scale-free neuronal avalanches. Although existing models account for oscillations and avalanches separately, they typically do not explain both phenomena, are too complex to analyze analytically or intractable to infer from data rigorously. Here we propose a feedback-driven Ising-like class of neural networks that captures avalanches and oscillations simultaneously and quantitatively. In the simplest yet fully microscopic model version, we can analytically compute the phase diagram and make direct contact with human brain resting-state activity recordings via tractable inference of the model's two essential parameters. The inferred model quantitatively captures the dynamics over a broad range of scales, from single sensor oscillations to collective behaviors of extreme events and neuronal avalanches. Importantly, the inferred parameters indicate that the co-existence of scale-specific (oscillations) and scale-free (avalanches) dynamics occurs close to a non-equilibrium critical point at the onset of self-sustained oscillations.

Synchronization is a key organizing principle that leads to the emergence of coherent macroscopic behaviors across diverse biological networks[1]. From Hebb's neural assemblies[2] to synfire chains[3], synchronization has also strongly shaped our understanding of brain dynamics and function[4]. The classic and arguably most prominent example of large-scale neural synchronization is brain oscillations, first reported about a century ago[5]: periodic, large deflections in electrophysiological recordings such as electroencephalography (EEG), magnetoencephalography (MEG) or local field potential (LFP)[4,5]. As oscillations are thought to play a fundamental role in brain function, their mechanistic origins have been the subject of intense research. According to the current view, the canonical circuit that generates prominent brain rhythms such as the alpha oscillations and the alternation of up- and down-states uses mutual coupling between excitatory (E) and inhibitory (I) neurons[6].

Alternative circuits, including I–I population coupling, have been proposed to explain other brain rhythms such as high-frequency gamma oscillations[7]. Setting biological details aside, the majority of research has predominantly focused on the emergence of synchronization at a preferred temporal scale—the oscillation frequency.

Yet brain activity also exhibits complex, large-scale cooperative dynamics with characteristics that are antithetic to those of oscillations. In particular, empirical observations of neuronal avalanches have shown that brain rhythms co-exist with activity cascades in which neuronal groups fire in patterns with no characteristic time or spatial scale, suggesting that the brain may operate near criticality[8–14]. In this context, the co-existence of scale-free neuronal avalanches with scale-specific oscillations suggests an intriguing dichotomy that is currently not understood. On the one hand, models of brain oscillations are

[1]Institute of Science and Technology Austria, Klosterneuburg, Austria. [2]Department of Cognitive and Brain Sciences, Ben-Gurion University of the Negev, Beer-Sheva, Israel. [3]Biofisika Institute (CSIC, UPV-EHU) and Ikerbasque Foundation, Bilbao, Spain. ✉e-mail: fabrizio.lombardi@ist.ac.at; gasper.tkacik@ist.ac.at; daniele.demartino@ehu.eus

very specific and seek to capture physiological mechanisms underlying particular brain rhythms. On the other hand, attempts to explain the emergence of neuronal avalanches almost exclusively focus on criticality-related aspects and ignore co-existing behaviors such as oscillations, even though they themselves may be constitutive for understanding the putative criticality. Among the few exceptions[15–19], Poil et al. proposed a probabilistic integrate and fire spiking model with E and I neurons, which generates long-range correlated fluctuations reminiscent of MEG oscillations in the resting state, with supra-threshold activity following power-law statistics consistent with neuronal avalanches and criticality[15]. More recently, by adopting a coarse-grained Landau–Ginzburg approach to neural network dynamics, Di Santo et al. have shown that neuronal avalanches and related putative signatures of criticality co-occur at a synchronization phase transition, where collective oscillations may also emerge[17]. These results were successively extended to a hybrid-type synchronization transition in a generalized Kuramoto model[20].

Although these and other proposed approaches show that neuronal avalanches may co-exist with some form of network oscillations[15,19] or network synchronization[17,20], they suffer from three major shortcomings. First, these models are neither simple (for example, in terms of parameters) nor analytically tractable, making an exhaustive exploration of their phase diagram out of reach. Second, neither of the two above-mentioned models simultaneously capture events at the microscopic (individual spikes) and macroscopic (collective variables) scales. Third, it is not clear how to rigorously connect these models to data, beyond relying on qualitative correspondences.

Here we propose a minimal, microscopic and analytically tractable model class that can capture a wide spectrum of emergent phenomena in brain dynamics, including neural oscillations, extreme event statistics and scale-free neuronal avalanches[8]. Inspired by recent theoretical results on the emergence of self-oscillations in systems with distinct co-existing phases[21], these models are non-equilibrium extensions of the Ising model of statistical physics with an extra feedback loop that enables self-adaptation. As a consequence of feedback, neuronal dynamics is driven by the ongoing network activity, generating a rich repertoire of dynamical behaviors. The structure of the simplest model from this class permits microscopic network dynamics investigations as well as an analytical mean-field solution in the Laudau–Ginzburg spirit and, in particular, allows us to construct the model's phase diagram.

The tractability of our model enables us to make direct contact with MEG data on the resting-state activity of the human brain. With its two free parameters inferred from data, the model closely captures brain dynamics across scales, from single sensor MEG signals to collective behavior of extreme events and neuronal avalanches. Remarkably, the inferred parameters indicate that scale-specific (neural oscillations) and scale-free (neuronal avalanches) dynamics in brain activity co-exist close to a non-equilibrium critical point that we proceed to characterize in detail.

## Results

### Adaptive Ising model
We consider a population of interacting neurons whose dynamics is self-regulated by a time-varying field that depends on the ongoing population activity level (Fig. 1a). The $N$ spins $s_i = \pm 1$ ($i = 1, 2, \ldots, N; N = 10^4$ in our simulations unless specified otherwise) represent excitatory neurons that are active when $s_i = +1$ or inactive when $s_i = -1$. In the simplest, fully homogeneous scenario described here, neurons interact with each other through synapses of equal strength $J_{ij} = J = 1$ (Methods).

The ongoing network activity is defined as $m(t) = \frac{1}{N}\sum_{i=1}^{N} s_i(t)$ (that is, as the magnetization of the Ising model) and each neuron experiences a uniform negative feedback $h$ that depends on the network activity as $\dot{h} = -cm$, with $c$ determining the strength of the feedback. Neurons $s_i$ are stochastically activated according to Glauber dynamics, where

the new state of neuron $s_i$ is drawn from the marginal Boltzmann–Gibbs distribution $P(s_i) \propto \exp(\beta\tilde{h}_i s_i)$, with $\tilde{h}_i = \sum_{j\neq i} J_{ij}s_j + h$, where $\beta$ is reminiscent of the inverse temperature for an Ising model (see Methods).

Multiple interpretations of this model are possible. On the one hand, negative feedback can be identified with a mean-field approximation to the inhibitory neuron population that uniformly affects all excitatory neurons with a delay given by the characteristic time $c^{-1}$ (Supplementary Section 1.4). On the other hand, feedback could be seen as intrinsic to excitatory neurons, mimicking, for example, spike-threshold adaptation[22]. Exploration-worthy (and possibly more realistic) extensions within the same model class are accessible by considering two ways in which geometry and neural biology can enter the model. First, as in the standard Ising magnet, the interaction matrix $J_{ij}$ can be used to model cell types (for example, inhibitory versus excitatory types; Supplementary Section 1.4), the spatial structure of the cortex, or empirically established topological features of real neural networks (Supplementary Section 1.5). Second, feedback $h_i$ to neuron $i$ could be derived from a local magnetization in a neighborhood around neuron $i$ instead of the global magnetization; in the interesting limiting case in which $\dot{h}_i = -cs_i$, each neuron would feedback only on its own past spiking history and the model would reduce to a set of coupled binary oscillators (see Supplementary Section 1.2 and Supplementary Fig. 2 for a discussion of this limiting case). Irrespective of the exact setting, the model's mathematical attractiveness stems from its tractable interpolation between stochastic (spiking of excitatory units) and deterministic (feedback) elements.

Here we consider the fully connected continuous time limit of the model (Fig. 1 and Methods). Network behavior is determined by $c$ and $\beta$. For $c = 0$, $h = 0$, the model reduces to the standard infinite-dimensional (mean field) Ising model with a second-order phase transition at $\beta = \beta_c = 1$. At non-zero feedback, $c > 0$, the model is driven out of equilibrium and its critical point at $\beta_c$ coincides with an Andronov–Hopf bifurcation[21]. For $\beta < \beta_c$ and $c$ below a threshold value $c^* = (\beta-1)^2/4\beta$, $m(t)$ is described by an Ornstein–Uhlenbeck process independently of $\beta$. For $\beta < \beta_c$, the system is stable and shows a cross-over from a stable node with exponential relaxation (two negative real eigenvalues) to a stable focus with oscillation-modulated exponential relaxation (two complex eigenvalues; resonant regime) when $c$ increases beyond $c^*$ (Supplementary Fig. 7). In the resonant regime, $c > c^*$, oscillations become more prominent as $\beta_c = 1$ is approached, finally transitioning into self-sustained oscillations for $\beta > \beta_c$ (Supplementary Fig. 8). At $\beta = \beta_c$, we have a line of Andronov–Hopf bifurcations where a focus loses stability and a limit cycle emerges. We find that this bifurcation is supercritical, with the first Lyapunov coefficient being negative (that is, $\lambda_1 = -(1+c)\beta/8 < 0$).

We focus on the resonant regime below and at the critical point, and study the reversal times and zero-crossing areas of the total network activity $m(t)$ (Fig. 1c). The reversal time, $t$, is defined as the time interval between two consecutive points in time at which a given signal crosses zero. Correspondingly, the zero-crossing area ($a_0$) is the area under the signal curve between two zero-crossing points. The distribution $P(a_0)$ of the zero-crossing area follows a power-law behavior with an exponent $\tau = 1.227 \pm 0.004$ in the vicinity of the critical point. As $\beta$ decreases, the scaling regime shrinks until it eventually vanishes for small enough $\beta$. Similar behavior is observed for the distribution $P(t)$ of reversal times. This distribution also follows a power-law with an exponent $\alpha_t = 1.378 \pm 0.004$ near the critical point (Fig. 1d). Both distributions have an exponential cutoff related to the characteristic time of the network activity oscillations, $1/c$; this cutoff transforms into a hump as $\beta \to 1$ and $c \gg c^*(\beta)$, that is, as oscillations in $m(t)$ become increasingly prominent (Supplementary Fig. 9). Importantly, for the non-interacting ($J = 0$) model, the distributions $P(a_0)$ and $P(t)$ follow a purely exponential behavior (Fig. 1d, inset), indicating that the co-existence of oscillatory bursts and power-law distributions for the network activity requires neuron interactions as well as the adaptive feedback (Supplementary Fig. 10).

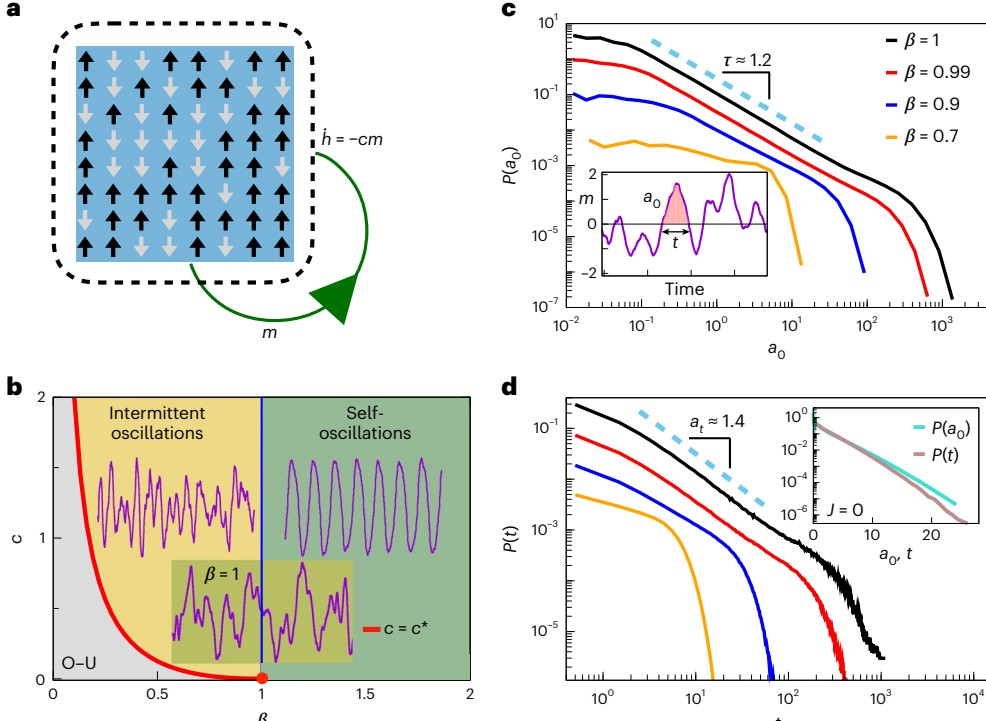

**Fig. 1 | Co-existence of oscillations and scale-free activity excursions in the adaptive Ising model near the critical point. a**, Schematic illustration of the model. Interacting spins $s_i$ ($i = 1, 2, …, N$) take values of +1 (up arrows) or −1 (down arrows), and experience a time-varying external field $h(t)$ that mimics an activity-dependent feedback mechanism. In the fully connected continuous time limit, the system can be described by the coupled Langevin equations $\dot{m} = -m + \tanh\left[\beta(Jm + h)\right] + b\xi$ and $\dot{h} = -cm$, where $\xi$ is unit-uncorrelated Gaussian noise and $b = \sqrt{2/(\beta N)}$. These equations can be linearized around the stationary point ($m^* = 0$, $h^* = 0$) to calculate dynamical eigenvalues and construct a phase diagram. **b**, Phase diagram for the mean-field adaptive Ising model. An Andronov–Hopf bifurcation at $\beta_c = 1$ separates self-sustained oscillations in $m(t)$ for $\beta > \beta_c$ (green shading) from the regime of intermittent oscillations for $c$ above $c^*(\beta)$ (yellow shading) and an Ornstein–Uhlenbeck process (O–U) for $c$ below $c^*$

(gray shading). **c**, The reversal time $t$ is the time interval between consecutive zero-crossing events in $m$, whereas $a_0$ is the area under the $m(t)$ curve between two zero-crossing events (inset). Distributions $P(a_0)$ are shown in the resonant regime $c > c^*$ for different values of $\beta$. When $\beta \approx 1$, $P(a_0)$ is approximately power-law with exponent $\tau = 1.227 \pm 0.004$; $\tau$ was estimated in the $a_0 \in [0.1, 100]$ range. **d**, Distributions $P(t)$ of the reversal times are shown in the resonant regime $c > c^*$ for different values of $\beta$. When $\beta \approx 1$, $P(t)$ is approximately power-law with exponent $\alpha_t = 1.378 \pm 0.004$; $\alpha_t$ was estimated in the $t \in [2, 500]$ range. The inset shows distributions $P(a_0)$ and $P(t)$ for the uncoupled model $J = 0$, which always exhibit exponential instead of power-law behavior (note the linear horizontal scale). Power-law fits were performed using a maximum likelihood estimator (Supplementary Section 1.8).

## Model inference from local resting-state brain dynamics

In the resonant regime below the critical point ($c > c^*$, $\beta < \beta_c$), it is possible to analytically compute the autocorrelation function, $C(\tau)$, of $m(t)$ in the linear approximation[23] (Methods); $C(\tau)$ can be used to infer model parameters $\beta$ and $c$ from empirical data by moment matching (see Supplementary Section 1.6 for details on parameter inference), thereby locating the observed system in the phase diagram (Fig. 1b).

We test the proposed approach on MEG recordings of the awake resting-state of the human brain (Methods). We first analyze brain activity on individual MEG sensors. To this end, we compare the magnetic field recorded on individual MEG sensors with the magnetization $m$ of the model (Fig. 1). This analogy relies on the nature of the brain magnetic fields captured by the MEG, which are generated by synchronous post-synaptic currents in cortical neurons, and on their relationship with collective neural fluctuations mimicked by $m$ (ref. [24]).

During resting wakefulness the brain activity is largely dominated by oscillations in the alpha band (8–13 Hz; Fig. 2a), which have been the starting point of many investigations[4,25,26] including ours reported below; similar results are also obtained for the broadband activity (Supplementary Fig. 11). After isolating the alpha band, we estimate $\beta$ and $c$ by fitting the empirical $C(\tau)$ to the analytical form of the autocorrelation (Methods). Figure 2b illustrates the typical quality of the fit

and the qualitative resemblance between the model and MEG sensor signal dynamics.

As our model is fit to reproduce the second-order statistical structure in the signal, we next turn our attention to signal excursions over the threshold − $e$, a higher-order statistical feature routinely used to characterize bursting brain dynamics[10,27–29]. To that end, we construct the distribution of (log) areas under the signal above a threshold $\pm e$ (Fig. 2c)[26]; $P(\log a_e)$ is bell-shaped, featuring strongly asymmetric tails for MEG sensors as well as the model (Fig. 2c). Variability across subjects is mostly related to signal amplitude modulation, resulting in small horizontal shifts in $P(\log a_e)$ but no variability in the distribution shape. Importantly, the rescaled distribution is independent of the threshold $e$ over a robust range of values, and is well-described by a Weibull form,

$$P_W(x; \lambda, k) = \frac{k}{\lambda}\left(\frac{x}{\lambda}\right)^{k-1} e^{-(x/\lambda)^k}$$ (Fig. 2c, bottom panel inset; Supplementary

Fig. 12). Taken together, these observations indicate that our model has the ability to capture non-trivial aspects of amplitude statistics in MEG signals, within and across different subjects (Supplementary Fig. 13).

Parameters inferred across all sensors and subjects suggest baseline values of $\beta = 0.99$ and $c = 0.01$ that are well matched with the data, which we use for all subsequent analyses (unless stated otherwise). Specifically, we find that the best-fit $\beta$ values strongly concentrate in a narrow range around $\beta \approx 0.99$ ($\beta = 0.986 \pm 0.006$; $c = 0.012 \pm 0.001$),

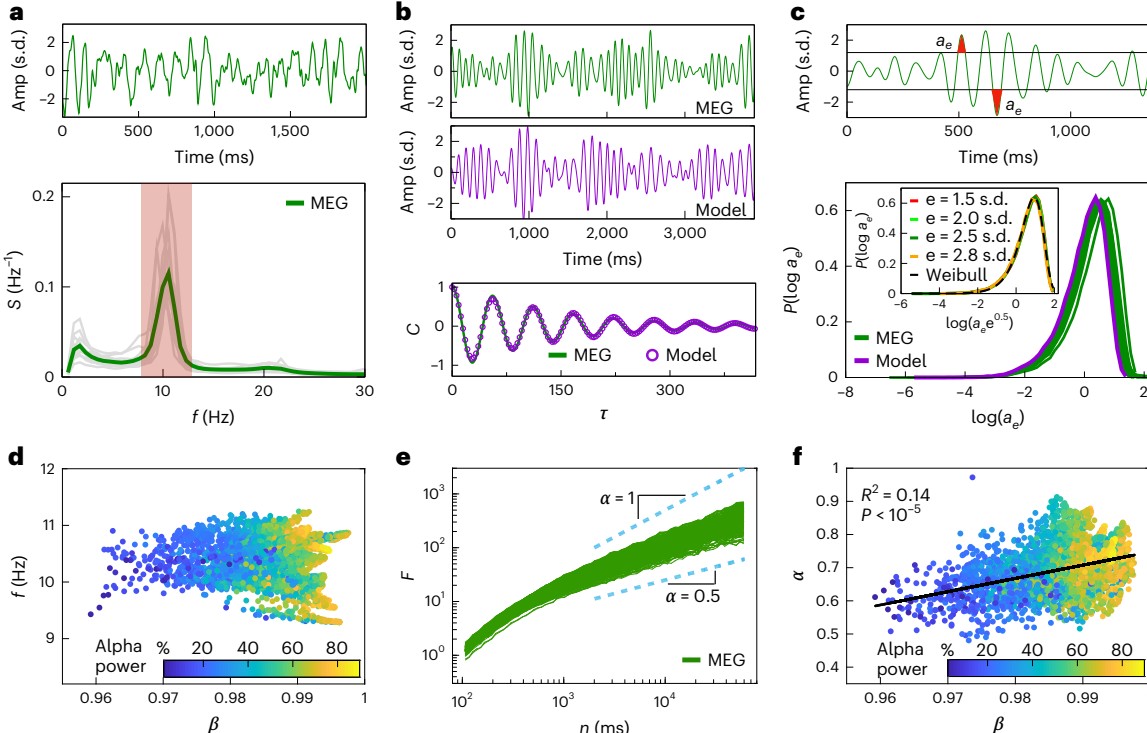

**Fig. 2 | Correspondence between MEG resting-state activity of the human brain and a marginally subcritical adaptive Ising model. a**, Example trace from a single MEG sensor (top) that predominantly contains power in the alpha band (8–13 Hz; bottom, red shaded region). The power spectra of MEG signals peak at around 10 Hz (bottom, the gray curve represents the average across 273 MEG sensors for each of the 14 subjects, whereas the green curve represents the average over sensors and subjects). Amp., amplitude of the MEG signal. **b**, Example of an alpha bandpass-filtered MEG signal (top, green curve) and the simulated $m(t)$ of a model with parameters matched to the data (top, violet curve). The model parameters ($\beta = 0.9870$ and $c = 0.0113$ for this trace) are inferred by fitting the analytical form of $C(\tau)$ (bottom, violet dots) to autocorrelation estimated from MEG data (bottom, green line; typical standard error of a $C(\tau)$ estimate $\approx 10^{-3}$). **c**, Top: schematic of the area under the curve $a_e$ (red shaded area) for a given threshold $\pm e$ in units of signal s.d. Bottom: distributions $P(\log a_e)$ of the logarithm of $a_e$ (with $e = 2.5$ s.d.) for MEG data (each

dark green curve represents the average over the sensors for each subject) and the model (the violet curve represents the simulation at baseline parameters; see main text). The inset shows that rescaled distributions of $a_e$ collapse to a universal Weibull-like distribution across different values of $e$ (Weibull parameters: $k = 1.74$, $\lambda = 2.58$). **d**, Central frequency $f = \omega/2\pi = (\beta c - (1 - \beta)^2)^{1/2}/8\pi$ of the fitted model plotted against fitted $\beta$, across all MEG sensors and subjects (the colors represent the fraction of the total MEG signal power in the alpha band); $\beta$ values closer to $\beta_c = 1$ are correlated with higher power in the alpha band (linear model fit $y = ax + b$; $R^2 = 0.21$; $P = 2 \times 10^{-193}$; Supplementary Section 1.8). **e**, Root-mean-square fluctuation function $F(n)$ of the DFA for the amplitude envelope of MEG sensor signals in the alpha band (the green lines represent individual sensors for a single subject); $F(n)$ scales as $F(n) \propto n^\alpha$ for $2\,s < n < 60\,s$ (light blue dashed lines), with $0.53 < \alpha < 0.85$. **f**, Inferred $\beta$ values correlate with the corresponding DFA exponents $\alpha$ for all MEG sensors and subjects (linear model fit $y = ax + b$; $P = 3 \times 10^{-131}$; Supplementary Section 1.8).

which is very close to the critical point (Fig. 2d and Supplementary Fig. 14). Although all analyzed signals are bandpass-limited to a central frequency of around 10 Hz by filtering, closeness to criticality seems to strongly correlate with the fraction of the total power in the raw signal in the alpha band (Fig. 2d; $R^2 = 0.21$; $P < 10^{-5}$). This suggests that alpha oscillations may be closely related to critical brain tuning during the resting state[11,25,30].

A classic fingerprint of tuning to criticality is the emergence of long-range temporal correlations (LRTCs), which have been documented empirically[25,29,30]. Long-range temporal correlations in the alpha band have been investigated primarily by applying the detrended fluctuations analysis (DFA) to the amplitude envelope of MEG or EEG signals in the alpha band (Methods)[15,25]. Briefly, DFA estimates the scaling exponent $\alpha$ of the root-mean-square fluctuation function $F$ in non-stationary signals with polynomial trends[31]. In brief, the integrated signal is divided into windows of equal length, $n$, and the local trend is subtracted in each window. For signals exhibiting positive (or negative) LRTC, $F$ scales as $F \propto n^\alpha$ with $0.5 < \alpha < 1$ (or $0 < \alpha < 0.5$); $\alpha = 0.5$ indicates the absence of long-range correlations; α also approaches unity for a number of known model systems as they are tuned to criticality[32].

To test for the presence of LRTC using DFA, we analyzed the scaling behavior of fluctuations and extracted their scaling exponent $\alpha$.

To avoid spurious correlations introduced by signal filtering, $\alpha$ was estimated over the range $2\,s < n < 60\,s$ (Fig. 2e)[25]. We find that $\alpha$ is consistently between 0.5 and 1 for all MEG sensors and subjects, in agreement with previous analyses[25]. Importantly, model-free $\alpha$ values measured across MEG sensors positively correlate with the inferred $\beta$ values from the model (Fig. 2f), indicating that higher $\beta$ values are diagnostic of the presence of long-range temporal correlations in the amplitude envelope. Furthermore, we find that inferred $\beta$ values correlate with the fraction of total signal power in the alpha band (Fig. 2d), which in turn correlates with the inferred entropy production in brain signals (Supplementary Section 1.1)[33].

Taken together, our analyses so far show that the adaptive Ising model recapitulates single-MEG-sensor dynamics by matching their autocorrelation function and the distribution of amplitude fluctuations, and further suggest that the true MEG signals are best reproduced when the adaptive Ising model is tuned close to, but slightly below, its critical point ($\beta \lesssim 1$).

## Scale-invariant collective dynamics of extreme events

We now turn our attention to phenomena that are intrinsically collective: (1) coordinated supra-threshold bursts of activity, which emerge

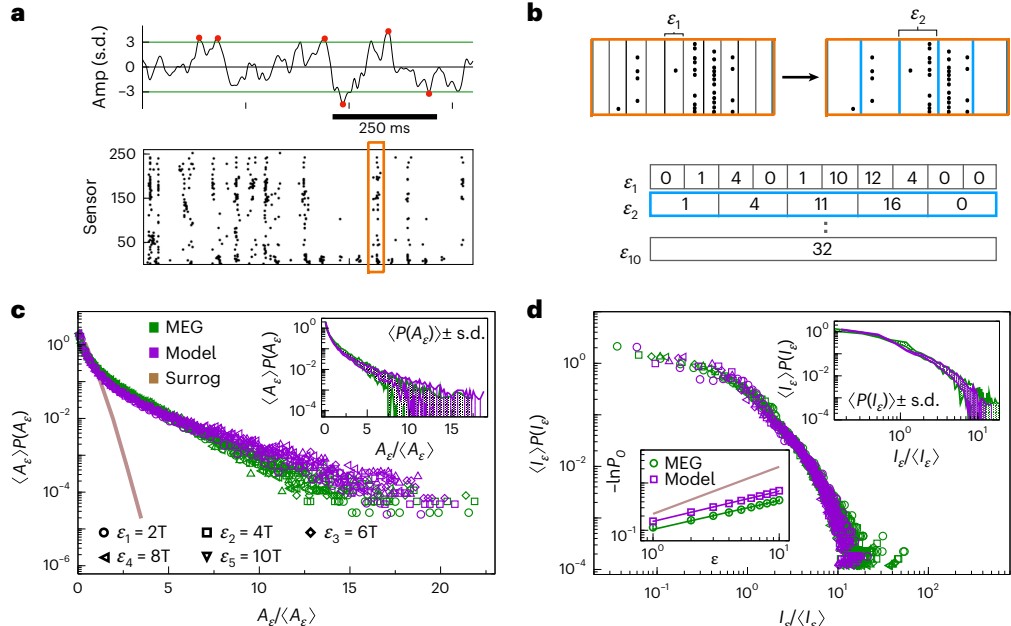

**Fig. 3 | Connecting non-exponential extreme event statistics in MEG resting-state activity and in a marginally subcritical adaptive Ising model.**
**a**, Top: extreme events identified on a single sensor (red dots) using $e = \pm 3$ s.d. (horizontal lines). Bottom: the resulting raster of extreme events shown across 273 MEG sensors of a single subject. **b**, Top: $A_\epsilon$ is defined as the total number of extreme events across all sensors in a time bin $\epsilon_n = nT$, a multiple of the sampling interval $T$. Bottom: representative sequences of network excitation extracted from the raster in the top panel for increasing $\epsilon_n$. **c**, Rescaled distributions $P(A_\epsilon)$ for a range of $\epsilon_n$ (different plot symbols) collapse onto a single non-exponential master curve for both the data (green symbols represent the average over subjects) and model simulated at baseline parameters, with $K = 100$ subsystems of $n_{sub} = 1,000$ neurons each (violet symbols) ($N = 10^5$). The corresponding distribution in phase-scrambled MEG signals shows an exponential behavior, with absence of high excitation events (the brown line represents surrogate

data). Inset: rescaled $P(A_\epsilon)$ (green symbols represent the average over $n = 14$ MEG subjects, whereas violet symbols represent the average over the model simulations) and respective s.d. (colored areas) shown for $\epsilon = 2T$. **d**, Rescaled distributions of quiescence durations, $P(I_\epsilon)$ collapse onto a single master curve for different values of $\epsilon$. The plotting conventions and model simulation details are the same as in **c**. Top inset: rescaled $P(I_\epsilon)$ (green symbols represent the average of $n = 14$ MEG subjects, whereas violet symbols are the average over model simulations) and respective s.d. (colored area) shown for $\epsilon = 2T$. Bottom inset: the probability $P_0$ of finding a quiescent time bin approximately scales as $P_0 = \exp\left(-r_0\epsilon^{\beta_I}\right)$ with $\epsilon$; $\beta_I = 0.582 \pm 0.013$ and $\beta_I = 0.610 \pm 0.012$ for the data and model, respectively; $\beta_I = 0.996 \pm 0.001$ for surrogate data. The exponent $\beta_I$ was estimated via an ordinary linear least-square fit $y = ax + b$, where $y = \ln(-\ln P_0)$ and $x = \ln(\epsilon)$.

jointly with LRTC in alpha oscillations[15]; and (2) neuronal avalanches, that is, spatio-temporal cascades of threshold-crossing sensor activity, which have been identified in the MEG of the resting state of the human brain[11,30]. Both of these phenomena are generally seen as chains of extreme events that are diagnostic of the underlying brain dynamics[10,34].

We start by defining the instantaneous network excitation $A_\epsilon(t)$ as the number of extreme events co-occurring within time bins of size $\epsilon$ across the entire MEG sensor array (Methods). For each sensor, extreme events are the extreme points in that sensor's signal that exceed a set threshold $e = \pm n$ s.d. (Fig. 3a). For a given threshold, $A_\epsilon$ depends on the size of the time bin $\epsilon$ that we use to analyze the data (Fig. 3b). To make contact with the model, we parcel our simulated network into $K$ equally sized disjoint subsystems of $n_{sub} = N/K$ neurons each, and consider each subsystem activity $m_\mu$ ($\mu = 1, \ldots, K$) as the equivalent of a single MEG sensor signal (Methods); $A_\epsilon$ for the model then follows the same definition as for the data, allowing us to perform direct side-by-side comparisons of extreme event statistics.

We first study the distribution of the network excitation, $P(A_\epsilon)$. We use the same threshold value $e = 2.9$ s.d. for both the data and model analyses (see Methods). Extensive robustness analyses confirm that our key results are stable in the 2.7 s.d. $< e < 3.1$ s.d. range (Supplementary Figs. 19 and 20), which we detail result-by-result below.

Although $P(A_\epsilon)$ generally depends on $\epsilon$, the distributions corresponding to different $\epsilon$ collapse onto a single, non-exponential master curve when $A_\epsilon$ is rescaled by the average instantaneous network excitation $\langle A_\epsilon \rangle$ (Fig. 3c). The excitation distribution is thus invariant

under temporal coarse-graining and the number of extreme events scales non-trivially with $\epsilon$, in contrast to phase-shuffled surrogate data (Methods and Fig. 3c). Model simulations fully recapitulate this data collapse as well as the non-exponential extreme event statistics. Moreover, we show that model simulations reproduce $P(A_\epsilon)$ to within the variability observed among subjects (Fig. 3c, inset) for given values of $\epsilon$. An analysis of the Kullback–Leibler divergence (Supplementary Section 2) shows that the model quantitatively reproduces the measured distributions to an expected degree given the natural variability in the data (Supplementary Table 1).

Periods of excitation ($A_\epsilon \neq 0$) are separated by periods of quiescence ($A_\epsilon = 0$) of duration $I_\epsilon = n\epsilon$, where $n$ is the number of consecutive time bins with $A_\epsilon = 0$. The distribution of quiescence durations, $P(I_\epsilon)$, is invariant under temporal coarse-graining when rescaled by the average quiescence duration, $\langle I_\epsilon \rangle$, collapsing onto a single, non-exponential master curve (Fig. 3d). As was the case with the distribution of network excitation, the model-predicted distribution of quiescence durations also diverges from the data average distribution by an amount that is within the range of variability among subjects (Fig. 3d, upper inset and Supplementary Table 1).

We also show that the overall probability $P_0(\epsilon)$ of finding a quiescent time bin follows a non-exponential relation $P_0(\epsilon) = \exp\left(-r_0\epsilon^{\beta_I}\right)$, with $\beta_I \simeq 0.6$ (Fig. 3d, lower inset), indicating that extreme events grouped into bins of increasing size are not independent[35]. These results are robust to changes in $N$, so long as $n_{sub}$ or the number of subsystems $K$ is fixed, or does not change considerably (Supplementary

Figs. 15 and 16); otherwise, the value of $e$ that defines an extreme event should be adjusted accordingly, in particular to closely reproduce the distribution of $P(I_\epsilon)$ (Supplementary Fig. 17). Finally we notice that the quantities $\langle A_\epsilon \rangle$ and $\langle I_\epsilon \rangle$ scale as a power of the bin size $\epsilon$ (Supplementary Fig. 21), and are connected to each other by a relationship of the form $\langle A_\epsilon \rangle \sim \langle I_\epsilon \rangle^{b_{AI}}$ (Supplementary Fig. 21). This implies that for a fixed value of $e$, both distributions $P(A_\epsilon)$ and $P(I_\epsilon)$ are controlled by a single quantity, for example, $\langle A_\epsilon \rangle$.

We performed the data and model analyses using the same threshold value $e = 2.9$ s.d., which was fixed by comparing the amplitude distribution of MEG sensor signals and model subsystem signals $m_\mu$. The distributions $P(A_\epsilon)$ and $P(I_\epsilon)$ follow a similar functional behavior in both the data and model for different values of $e$. The influence of thresholding on the analysis of continuous signals has been previously investigated[36]. Here, for increasing values of $e$, we find that: (1) the probability of large (small) $A_\epsilon$ tends to decrease (increase); (2) the probability of large (small) $I_\epsilon$ tends to increase (decrease) (Supplementary Fig. 18). These effects are more pronounced for the distribution of $P(I_\epsilon)$, particularly in its tail. Importantly, $P(A_\epsilon)$ and $P(I_\epsilon)$, as well as the exponent $\beta_I$, show a similar dependence on $e$ in both MEG data and model simulations and, as a consequence, the agreement between the data and model is robust to changes in $e$ (Supplementary Figs. 19 and 20).

In summary, our simple model at baseline parameters provides a robust account of the collective statistics of extreme events. We emphasize that the excellent match to the observed long-tailed distributions is only observed for the inferred value $\beta \simeq 0.99$, which is very close to criticality; for $\beta = 0.98$, we already observe considerable deviations from the data (Supplementary Figs. 22 and 23), demonstrating that excitation and quiescence distributions represent a powerful benchmark for collective brain activity.

## Concomitant occurrence of scale-free neuronal avalanches and scale-specific oscillations

A neuronal avalanche is a maximal contiguous sequence of time bins populated with at least one extreme event per bin (Fig. 4a)[8,11]; every avalanche thus starts after—and ends with—a quiescent time bin ($A_\epsilon = 0$) (see Methods for details). Neuronal avalanches are typically characterized by their size $s$, defined as the total number of extreme events within the avalanche. Avalanche sizes have been reported to have a scale-free power-law distribution[8,11,14,30].

We estimate the distribution of avalanche sizes $P(s)$ in the resting-state MEG, and compare it with the distribution obtained from model simulation at close-to-critical baseline parameter set (Fig. 4b). Both distributions are described by a power-law with an exponential cut-off[11] and show an excellent match across subjects and for individual subjects. Again, the Kullback–Leibler divergence between the mean empirical and model distribution is smaller than the mean Kullback–Leibler divergence estimated among MEG subjects (Supplementary Table 1). Phase-scrambled surrogate data strongly deviate from the power-law observations, as do model predictions when parameter $\beta$ is moved even marginally below 0.99 (Supplementary Fig. 24). These results are independent of the $N$ so long as the size $n_{sub}$ or the number $K$ of the subsystems are fixed or do not change considerably (Supplementary Figs. 15 and 16). Importantly, the model also reproduces the distribution of avalanche durations (Supplementary Fig. 26) and, in particular, the scaling relation $\langle s \rangle(d) \sim d^\zeta$ that connects average avalanche sizes $s$ and durations $d$. Unlike the power-law exponent of avalanche size distribution that typically depends on time bin size $\epsilon$ (refs. [8,30]), the exponent $\zeta$ does not depend on $\epsilon$, as shown by the data collapse for both MEG data and model (Fig. 4b, inset). Although the scaling behavior is reproduced qualitatively, the inferred and model-derived values of $\zeta$ are not in quantitative agreement, probably due to the overly simplified mean-field connectivity assumed by our model.

As shown for $P(A_\epsilon)$ and $P(I_\epsilon)$, the distributions of avalanche sizes also moderately depend on $e$. This has been previously reported both

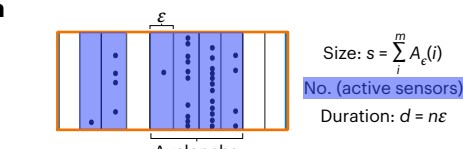

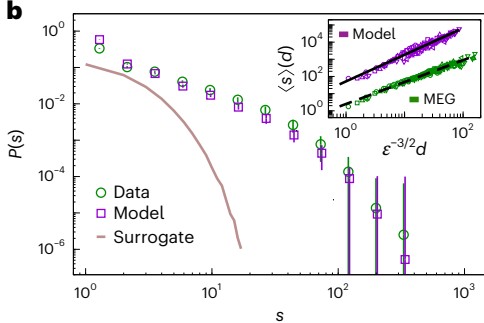

**Fig. 4 | Reproducing scale-free neuronal avalanches in MEG resting-state activity with a marginally subcritical adaptive Ising model. a**, Schematic representation of a neuronal avalanche. The avalanche size $s$ is the sum of network excitations $A_\epsilon$ over time bins belonging to the avalanche; its duration, $d$, is the number of bins multiplied by their duration, $\epsilon$. **b**, Distribution of avalanche sizes, $P(s)$, for MEG data (the green circles with error bars represent the average over the subjects ± s.d.) and the model simulated at baseline parameters with $K = 100$ subsystems of $n_{sub} = 100$ neurons each (the violet squares with error bars represent the average over the model simulations ± s.d.). Both distributions are estimated using $e = 2.9$ s.d. and $\epsilon_4 = 4T$. The brown curve represents the $P(s)$ obtained from the surrogate data (Supplementary Section 1.8) with the same threshold and bin size. The inset shows that the average avalanche size scales with its duration as $\langle s \rangle \sim d^\zeta$ (different plot symbols represent different $\epsilon$, as in Fig. 3; green, MEG data; violet, model simulation; model simulation curves are vertically shifted for clarity) so that the exponent $\zeta$ remains independent of $\epsilon$; $\zeta = 1.28 \pm 0.01$ for the MEG data (dashed line) and $\zeta = 1.58 \pm 0.03$ for the model simulation (thick line). The exponent $\zeta$ was estimated via ordinary linear least-square fit $y = ax + b$, with $y = \ln\langle s \rangle$ and $x = \ln(\epsilon^{-3/2}d)$.

in the resting human brain and in other systems[8,30]. We find that simulated avalanche size distributions show a similar dependence to the data, and are thus in agreement with empirical distributions for a range of $e$ values (Supplementary Fig. 25). Importantly, we observe that the relationship between avalanche sizes and durations is robust to changes in $e$, and the exponent $\zeta$ shows no substantial dependence on $e$ (Supplementary Figs. 18 and 25).

## Discussion

In this paper we put forward the adaptive Ising class of models for capturing large-scale brain dynamics. To our knowledge, this is the simplest model class that reproduces the stylized co-existence of neuronal avalanches and oscillations—the two antithetic features of real brain dynamics. In this formulation, individual units are neither intrinsic oscillators themselves[20,37], nor are they mesoscopic units operating close to a Hopf bifurcation[38]; the collective dynamics is therefore not a result of oscillator synchronization (even though this regime could also be captured by a different realization of an adaptive Ising model). Our proposal thus provides an analytically tractable alternative to, or perhaps a reformulation of, existing models[15,17,19,39], which typically implicate either particular excitation/inhibition or network resource balance, or ad hoc driving mechanisms to open up the regime in which oscillations and avalanches may co-exist.

Starting with the seminal work of Hopfield[40], the functional aspects of neural networks have traditionally been studied with microscopic spin models or attractor neural networks. The associated inverse (maximum entropy) problem recently attracted great attention in connecting spin models to data[41,42], particularly with regards to criticality

signatures[43] and the structure of temporal correlations in the neural activity[44,45]. However, the dynamical expressive power of maximum-entropy stationary, kinetic or latent-variable models has been limited, and the rhythmic behavior of brain oscillations was beyond the practical scope of these models. The adaptive Ising model class can be seen as a natural yet orthogonal extension to those previous works, as it enables oscillations and furthermore permits us to explore an interesting interplay of mechanisms, for example, by having self-feedback drive Hopfield-like networks (with memories encoded in the coupling matrix $J$) through sequences of stable states.

By contrast to past works[15,17], we do not make contact with existing data by qualitatively matching the phenomenology, but instead by proper parameter inference. The inferred parameters consistently place the model very close to its critical point, supporting the hypothesis that alpha oscillations represent brain tuning to criticality[25]. Inference of parameters with methods that are not based on autocorrelation matching[46] has confirmed this result (Supplementary Section 1.7 and Supplementary Figs. 3–6). Other models also predict adaptive parameters that are slightly subcritical[47]. However, within our framework, the possibility of mapping empirical data to a defined region in the adaptive Ising model phase diagram through parameter inference paves the way for further quantification of the relationship between measures of brain criticality and healthy, developing or pathological brain dynamics along the lines developed recently[48].

Our inferred model provides a broad account of brain dynamics across spatial and temporal scales. Despite the successes, we openly acknowledge the quantitative failures of our model: first, at the single sensor level, small deviations exist in the distributions of log activity (Fig. 2c), probably due to very long timescales or non-stationarities in the MEG signals[11]; second, the scaling exponent governing the relation between the avalanche size and duration, $\zeta$, is not reproduced quantitatively (Fig. 4b, inset). Despite these valid points of concern, we find it remarkable that such a simple and tractable model can quantitatively account for so much of the observed phenomenology.

Future work should, first, consider connectivity beyond the simple all-to-all mean-field version that we introduced here, probably leading to a better data fit and new types of dynamics, for example, cortical waves (Supplementary Section 1.5). Second, we strongly advocate for rigorous and transparent data analysis and quantitative—not only stylized—comparisons to data. To this end, care must be taken not only when inferring the essential model parameters beyond the linear approximation[46,49], but also when treating the hidden degrees of freedom related to the data analysis (specifically, subsampling, temporal discretization, thresholding and so on)[8,30,36,50]. Third, it is important to confront the model with different types of brain recordings; a real success in this vein would be to account simultaneously for the activity statistics at the microscale (spiking of individual neurons) as well as at the mesoscale (coarse-grained activity probed with MEG, EEG or LFP).

## Methods
### Data acquisition and preprocessing
Ongoing brain activity was recorded from 14 healthy participants in the MEG core facility at the National Institute of Mental Health for a duration of 4 min (eyes closed). All of the experiments were performed in accordance with the NIH guidelines for human subjects. All participants gave written informed consent. The sampling rate was 600 Hz, and the data were bandpass-filtered between 1 and 150 Hz. Power-line interferences were removed using a 60 Hz notch filter designed in Matlab (Mathworks). The sensor array consisted of 275 axial first-order gradiometers. Two dysfunctional sensors were removed, leaving 273 sensors in the analysis. The analysis was performed directly on the axial gradiometer waveforms. The data analyzed here were selected from a set of MEG recordings for a previously published study[11], in which further details can be found. For our analyses, we used the subjects

showing the highest percentage of spectral power in the alpha band (8–13 Hz). Similar results were obtained for randomly selected subjects.

### The adaptive Ising model
The model comprises a collection of $N$ spins $s_i = \pm 1$ ($i = 1, 2, \ldots, N$) that interact with each other with a coupling strength $J_{ij}$. In our analysis, the $N$ spins represent excitatory neurons that are active when $s_i = +1$ or inactive when $s_i = -1$ and $J_{ij} > 0$. Furthermore, we consider the fully homogeneous scenario in which neurons interact with each other through synapses of equal strength $J_{ij} = J = 1$. However, interesting generalizations with non-homogeneous, negative, non-symmetric $J_{ij}$ are possible, which allow to include in the model, for example, the effect of inhibitory neuronal population and structural and functional heterogeneity. The $s_i$ are stochastically activated according to the Glauber dynamics, where the state of a neuron is drawn from the marginal Boltzmann–Gibbs distribution

$$P(s_i) \propto \exp(\beta \tilde{h}_i s_i) \qquad \tilde{h}_i = \sum_j J_{ij} s_j + h_i. \qquad (1)$$

The spins experience an external field $h$, a negative feedback that depends on network activity according to the following equation,

$$\dot{h}_i = -c \frac{1}{\mathcal{N}_i} \sum_{j \in \mathcal{N}_i}^{|N_i|} s_j, \qquad (2)$$

where $c$ is a constant that controls the feedback strength, and the sum runs over a neighborhood of the neuron $i$ specified by $\mathcal{N}_i$; index $j$ enumerates over all of the elements of this neighborhood. Depending on the choice of $\mathcal{N}_i$, the feedback may depend on the activity of the neuron $i$ itself (self-feedback), its nearest neighbors, or the entire network—the case which we considered in the main paper. In a more realistic setting including both excitatory ($J_{ij} > 0$) and inhibitory neurons ($J_{ij} < 0$), one could then take into account the different structural and functional properties of excitatory and inhibitory neurons by considering different interaction and feedback properties.

In the fully connected continuous time limit, the model can be described with the following Langevin equations:

$$\begin{aligned} \dot{m} &= -m + \tanh\left[\beta(Jm + h)\right] + b\xi \\ \dot{h} &= -cm, \end{aligned} \qquad (3)$$

where $\xi$ is unit-uncorrelated Gaussian noise; the stochastic term thus has amplitude $b = \sqrt{2/(\beta N)}$. This framework allows for a reparametrizazion of spin variables $s_i$ from $(-1, 1)$ to $(0, 1)$ by introducing a constant term, $-cm_0$, in the feedback equation (Supplementary Section 1.3). Equation (3) can be linearized around the stationary point ($m^* = 0$, $h^* = 0$) to calculate dynamical eigenvalues and construct a phase diagram (Fig. 1b, main text):

$$\lambda_{\pm} = \frac{(\beta-1)}{2} \pm \frac{\sqrt{(\beta-1)^2 - 4c\beta}}{2}. \qquad (4)$$

In the resonant regime below the critical point ($c > c^*$, $\beta < \beta_c$), it is possible to analytically compute the autocorrelation function, $C(\tau)$, of the ongoing network activity $m(t)$ in the linear approximation[23]:

$$C(\tau) = e^{-\gamma\tau}(\cos \omega\tau + \frac{\gamma}{\omega} \sin \omega\tau), \qquad (5)$$

where $\gamma = (1 - \beta)/2$ is the relaxation time of the system, and $\omega = \sqrt{\beta c - (1 - \beta)^2/4}$ is the characteristic angular frequency of the model.

In our simulations, one time step corresponds to one system sweep—that is, $N$ spin flips—of Monte Carlo updates, and equation

([2]) is integrated using $\Delta t = 1/N$. Note that this choice of timescales for deterministic versus stochastic dynamic is important, as it interpolates between the quasi-equilibrium regime where spins fully equilibrate with respect to the field $h$, and the regime where the field is updated by feedback after each spin-flip and so spins can constantly remain out of equilibrium; $\Delta t$ is generally much smaller than the characteristic time of the adaptive feedback that is controlled by the parameter $c$.

### Detrended fluctuations analysis of the alpha band amplitude envelope

The DFA[31] consists of the following steps: (1) given a time-series $x_i (i = 1, \ldots, N)$, calculate the integrated signal $I(k) = \sum_{i=1}^{k}(x(i) - \langle x \rangle)$, where $\langle x \rangle$ is the mean of $x_i$; (2) divide the integrated signal $I(k)$ into boxes of equal length $n$ and, in each box, fit $I(k)$ with a first-order polynomial $I_n(k)$, which represents the trend in that box; (3) for each $n$, detrend $I(k)$ by subtracting the local trend, $I_n(k)$, in each box and calculate the root-mean-square (r.m.s.) fluctuation $F(n) = \sqrt{\sum_{k=1}^{N}[I(k) - I_n(k)]^2 / N}$; (4) repeat this calculation over a range of box lengths $n$ and obtain a functional relation between $F(n)$ and $n$. For a power-law correlated time-series, the average r.m.s. fluctuation function $F(n)$ and the box size $n$ are connected by a power-law relation $F(n) \approx n^\alpha$. The exponent $\alpha$ quantifies the long-range correlation properties of the signal. Values of $\alpha < 0.5$ indicate the presence of anti-correlations in the time-series $x_i$, $\alpha = 0.5$ indicates the absence of correlations (white noise), and values of $\alpha > 0.5$ indicate the presence of positive correlations in $x_i$. The DFA was applied to the alpha band (8–13 Hz) amplitude envelope. Data were band filtered in the 8–13 Hz range using a finite impulse response (FIR) filter (second order) designed in Matlab. The scaling exponent $\alpha$ was estimated in the $n$ range corresponding to 2–60 s to avoid spurious correlations induced by the signal filtering[25].

### Extreme events, instantaneous network excitation and neuronal avalanches

**Data.** For each sensor, positive and negative excursions beyond a threshold $e$ were identified. In each excursion beyond the threshold, a single event was identified at the most extreme value (the maximum for positive excursions and minimum for negative excursions). Comparison of the signal distribution with the best-fit Gaussian indicates that the two distributions start to deviate from one another at around $\pm 2.7$ s.d. (ref. [11]). A Gaussian distribution of amplitudes is expected to be produced from a superposition of uncorrelated sources, and is not indicative of individual extreme events. For such a reason, one needs to choose $e \geq 2.7$ s.d. for the threshold. Higher values will reduce the number of false positives, but increase the number of false negatives. In this study we set $e$ to $\pm 2.9$ s.d. We performed an extensive robustness analyses to confirm that our key results are stable across a range of $e$ values (Supplementary Figs. 19, 20 and 25).

The raster of identified events was binned at a number of temporal resolutions $\epsilon$, which are a multiple of the sampling time $T = 1.67$ ms. The network excitation $A_\epsilon$ at a given temporal resolution $\epsilon$ is defined as the number of events occurring across all sensors in a time bin. An avalanche is defined as a continuous sequence of time bins in which there is at least an event on any sensor, ending with at least a time bin with no events (Fig. 4a). The size of an avalanche, $s$, is defined as the number of events in the avalanche. See refs. [11,30] for more details.

**Model.** The simulated network is parceled into $K$ equally sized disjoint subsystems of $n_{sub} = N/K$ neurons each, and each subsystem activity $m_\mu (\mu = 1, \ldots, K)$ is considered as the equivalent of a single MEG sensor signal. The number of neurons $n_{sub}$ in each subsystem is fixed by matching the amplitude distribution of $m_\mu$ to the estimated MEG sensor amplitude distribution between $\pm 2.7$ s.d., which is the range over which amplitude distributions follow a Gaussian behavior[11]. This procedure gives the sufficient number of neurons whose collective activity

accounts for the the Gaussian core of the empirical signal amplitude distribution, thus providing a common reference to consistently define extreme events in empirical data and model simulations. Extreme events, network excitation and neuronal avalanches for the model follow the same definition as for the data.

**Data-model comparison.** Beyond the two key model parameters that are directly inferred from individual sensors ($\beta$, $c$), quantitative data analysis of extreme events requires additional parametric choices (time bin $\epsilon$, threshold $e$, system size $N$ and subsystem size $n_{sub}$), both for empirical data as well as model simulations. We successfully demonstrate the scaling invariance of the relevant distributions with respect to $\epsilon$, and robustness of results in a range of $e$ values (Supplementary Figs. 19, 20 and 25). Moreover, we demonstrate robustness with respect to $n_{sub}$ at fixed $K = N/n_{sub}$, and to $K$ at fixed $n_{sub}$. However, if $K$ (or $n_{sub}$) changes considerably, a close match to data (in particular, $P(I_\epsilon)$) still requires adjusting one extra parameter (for example, threshold $e$; Supplementary Fig. 17).

### Reporting summary

Further information on research design is available in the Nature Portfolio Reporting Summary linked to this article.

## Data availability

The data analyzed in this study were collected at the MEG facility of the NIH for a previously published study[11]. The data belong to NIH and are available from O.S. (shrikio@bgu.ac.il) on reasonable request. Source data are provided with this paper.

## Code availability

The codes[51] used in the current study are publicly available on GitHub (https://github.com/demartid/stat_mod_ada_nn).

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

## Acknowledgements

This research was funded in whole, or in part, by the Austrian Science Fund (FWF) (grant no. PT1013M03318 to F.L. and no. P34015 to G.T.). For the purpose of open access, the author has applied a CC BY public copyright licence to any Author Accepted Manuscript version arising from this submission. The study was supported by the European Union Horizon 2020 research and innovation program under the Marie Sklodowska-Curie action (grant agreement No. 754411 to F.L.).

## Author contributions

F.L., G.T. and D.D.M. designed the research and wrote the paper. F.L. and D.D.M. analyzed the data. All of the authors performed the research.

## Competing interest

The authors declare no competing interests.

## Additional information

**Correspondence and requests for materials** should be addressed to Fabrizio Lombardi, Gašper Tkačik or Daniele De Martino.

# Reporting Summary

## Statistics

For all statistical analyses, confirm that the following items are present in the figure legend, table legend, main text, or Methods section.

| n/a | Confirmed | |
|---|---|---|
| ☐ | ☒ | The exact sample size (*n*) for each experimental group/condition, given as a discrete number and unit of measurement |
| ☒ | ☐ | A statement on whether measurements were taken from distinct samples or whether the same sample was measured repeatedly |
| ☐ | ☒ | The statistical test(s) used AND whether they are one- or two-sided *Only common tests should be described solely by name; describe more complex techniques in the Methods section.* |
| ☒ | ☐ | A description of all covariates tested |
| ☒ | ☐ | A description of any assumptions or corrections, such as tests of normality and adjustment for multiple comparisons |
| ☐ | ☒ | A full description of the statistical parameters including central tendency (e.g. means) or other basic estimates (e.g. regression coefficient) AND variation (e.g. standard deviation) or associated estimates of uncertainty (e.g. confidence intervals) |
| ☐ | ☒ | For null hypothesis testing, the test statistic (e.g. *F*, *t*, *r*) with confidence intervals, effect sizes, degrees of freedom and *P* value noted *Give P values as exact values whenever suitable.* |
| ☒ | ☐ | For Bayesian analysis, information on the choice of priors and Markov chain Monte Carlo settings |
| ☒ | ☐ | For hierarchical and complex designs, identification of the appropriate level for tests and full reporting of outcomes |
| ☒ | ☐ | Estimates of effect sizes (e.g. Cohen's *d*, Pearson's *r*), indicating how they were calculated |

*Our web collection on statistics for biologists contains articles on many of the points above.*

## Software and code

Policy information about availability of computer code

| Data collection | The data was recorded using a CTF MEG system (CTF Systems). Data preprocessing was performed using the Field-Trip toolbox (version 2011-09) in MATLAB 2011a (Mathworks) |
|---|---|
| Data analysis | Data analysis was performed in MATLAB 2020b. |

For manuscripts utilizing custom algorithms or software that are central to the research but not yet described in published literature, software must be made available to editors and reviewers. We strongly encourage code deposition in a community repository (e.g. GitHub). See the Nature Portfolio guidelines for submitting code & software for further information.

## Data

Policy information about availability of data

All manuscripts must include a data availability statement. This statement should provide the following information, where applicable:
- Accession codes, unique identifiers, or web links for publicly available datasets
- A description of any restrictions on data availability
- For clinical datasets or third party data, please ensure that the statement adheres to our policy

The data analyzed in this study was collected at the MEG facility of the NIH. Data belongs to NIH and is available upon request. We have permission to use it and share it upon request. We do not have permission to upload the data into a public repository.

## Human research participants

Policy information about studies involving human research participants and Sex and Gender in Research.

| | |
|---|---|
| Reporting on sex and gender | Sex and gender were not relevant for the present study. We did not perform sex/gender based analyses. Although this information was collected for the original study (Shriki et al, J. Neurosci. 33 (16) : 7079 -7090, 2013; see Population characteristics), it was not accessible to the Authors and was not relevant for the selection of the 14 subjects used for the present study. Because we were interested in the connection between alpha oscillations and brain criticality, subject selection was solely based on the percentage of spectral power in the alpha band (8-13 Hz). |
| Population characteristics | The NIH facility recorded activity from 104 subjects (38 males and 66 females; age, 31.8 ± 11.8 ) for 4 min at rest with eyes closed (see Shriki et al, J. Neurosci. 33 (16) : 7079 -7090, 2013 for details). |
| Recruitment | Participants were healthy and had no history of neurological or psychiatric diseases. |
| Ethics oversight | NIH |

Note that full information on the approval of the study protocol must also be provided in the manuscript.

# Field-specific reporting

Please select the one below that is the best fit for your research. If you are not sure, read the appropriate sections before making your selection.

☒ Life sciences        ☐ Behavioural & social sciences        ☐ Ecological, evolutionary & environmental sciences

For a reference copy of the document with all sections, see nature.com/documents/nr-reporting-summary-flat.pdf

# Life sciences study design

All studies must disclose on these points even when the disclosure is negative.

| | |
|---|---|
| Sample size | The data analyzed here were selected from a set of MEG recordings for a previously published study (Shriki et al, J. Neurosci. 33 (16) : 7079 - 7090, 2013), where further details can be found. For the present analyses we used the subjects (n = 14) showing the highest percentage of spectral power in the alpha band (8-13 Hz) (the dominant brain rhythm during resting wake). Similar results were obtained for randomly selected subjects. Results are very consistent across subjects, and increasing the sample size does not significantly influence the results. The number of selected subjects is thus adequate to provides robust results. |
| Data exclusions | In Shriki et al, J. Neurosci. 33 (16) : 7079 -7090, 2013, data from 104 participants recorded in the NIHM MEG core facility were considered. Because we were interested in the connection between alpha oscillations and brain criticality, for the present analyses we used 14 subjects showing high percentage of spectral power in the alpha band (8-13 Hz). |
| Replication | Experimental findings were replicated in a different MEG facility at the Medical Research Council Cognition and Brain Sciences Unit in Cambridge (see Shriki et al, J. Neurosci. 33 (16): 7079 -7090, 2013 for details). Data analyzed in the current study were previously used in Shriki et al, J. Neurosci. 33 (16) : 7079 -7090, 2013. Experimental findings were replicated once and presented in Shriki et al. J. Neurosci. 33 (16) : 7079 -7090, 2013. |
| Randomization | We analyzed data recorded from a single group of healthy subjects. Only participants with no history of neurological or psychiatric diseases were selected for this study. Results were consistent across subjects. Thus randomization was not relevant for the present study. |
| Blinding | Only brain activity recorded from healthy subjects was analyzed in this study. The data analyzed here were selected from a set of MEG recordings for a previously published study (Shriki et al, J. Neurosci. 33 (16) : 7079-7090, 2013). Because we were interested in the connection between alpha oscillations and brain criticality, we used 14 subjects showing high percentage of spectral power in the alpha band (8-13 Hz). Similar results were obtained for randomly selected subjects. |

# Reporting for specific materials, systems and methods

We require information from authors about some types of materials, experimental systems and methods used in many studies. Here, indicate whether each material, system or method listed is relevant to your study. If you are not sure if a list item applies to your research, read the appropriate section before selecting a response.

## Materials & experimental systems

| n/a | Involved in the study |
|-----|----------------------|
| ☒ | ☐ Antibodies |
| ☒ | ☐ Eukaryotic cell lines |
| ☒ | ☐ Palaeontology and archaeology |
| ☒ | ☐ Animals and other organisms |
| ☒ | ☐ Clinical data |
| ☒ | ☐ Dual use research of concern |

## Methods

| n/a | Involved in the study |
|-----|----------------------|
| ☒ | ☐ ChIP-seq |
| ☒ | ☐ Flow cytometry |
| ☒ | ☐ MRI-based neuroimaging |

