## [Peer Review File · Nature Computational Science]

Peer Review Information

Journal: Nature Computational Science

Manuscript Title: Statistical modeling of adaptive neural networks explains coexistence of avalanches and oscillations in resting human brain

Corresponding author name(s): Daniele De Martino, Fabrizio Lombardi

Editorial Notes:

Reviewer Comments & Decisions:

Decision Letter, initial version:
--

Dear Dr Lombardi,

Your manuscript "Statistical modeling of adaptive neural networks explains coexistence of avalanches and oscillations in resting human brain" has now been seen by 3 referees, whose comments are appended below. You will see that while they find your work of interest, they have raised points that need to be addressed before we can make a decision on publication.

The referees' reports seem to be quite clear. Naturally, we will need you to address all of the points raised.

While we ask you to address all of the points raised, the following points need to be substantially worked on:

- As Reviewer #1 mentioned, the presented model is a new variation of the Ising model and therefore it is important to contrast previous models to the current one.
- Please discuss in the manuscript how the thresholds are chosen and what are the effects of choosing different thresholds.
- In the manuscript, many fits were carried out and the procedures for these fits should be discussed as mentioned by Reviewer #1.
- The adaptive Ising model's physical properties should be discussed.
- As mentioned by Reviewer #3, please discuss whether the inference procedure limits the inference to a restricted portion of the phase diagram and if it is possible to imagine a more general inference procedure?
- Please discuss how the size of the subsystems N/K is chosen to perform the "side-by-side" comparison with the data.

Please use the following link to submit your revised manuscript and a point-by-point response to the referees' comments (which should be in a separate document to any cover letter):

[REDACTED]

** This url links to your confidential homepage and associated information about manuscripts you may have submitted or be reviewing for us. If you wish to forward this e-mail to co-authors, please delete this link to your homepage first. **

To aid in the review process, we would appreciate it if you could also provide a copy of your manuscript files that indicates your revisions by making use of Track Changes or similar mark-up tools. Please also ensure that all correspondence is marked with your Nature Computational Science reference number in the subject line.

In addition, please make sure to upload a Word Document or LaTeX version of your text, to assist us in the editorial stage.

To improve transparency in authorship, we request that all authors identified as 'corresponding author' on published papers create and link their Open Researcher and Contributor Identifier (ORCID) with their account on the Manuscript Tracking System (MTS), prior to acceptance. ORCID helps the scientific community achieve unambiguous attribution of all scholarly contributions. You can create and link your ORCID from the home page of the MTS by clicking on 'Modify my Springer Nature account'. For more information please visit www.springernature.com/orcid.

We hope to receive your revised paper within three weeks. If you cannot send it within this time, please let us know.

Best regards,

Ananya Rastogi, PhD
Associate Editor
Nature Computational Science

Reviewers comments:

Reviewer #1 (Remarks to the Author):

The manuscript under review presents an application of the adaptive Ising model to neuroscience with analysis of MEG data. Both the model has been published before and the MEG data was also previously published in 2013. In my understanding, the original content of the paper consists of the matching of model parameters to the MEG data.

The work is both interesting and relevant, and certainly is worth publishing after some revisions, but I'm uncertain whether the content suffices for a publication in Nature Computational Science. My uncertainty is partially due to the fact that this journal is relatively new and it remains to be seen what level of novelty and impact is sought. I would certainly have concerns if the manuscript was submitted to Nature Physics or Nature Neuroscience.

Here are some major points that the authors should address:

- 1) The presented model is a new variation of the Ising model which has been applied to Neuroscience in many publications before. Because there have been many spin-like neuronal models, the bar for promoting or establishing a new variation of these models is high. It is important to contrast previous models to the current one: what have these models accounted for and what have they missed? How does the presented model perform better compared to previous ones? Some of this already becomes clear from the manuscript but I would recommend making a structured and comprehensive comparison. One important missing reference in this context is by C. Lynn et al. "Broken detailed balance and entropy production in the human brain", PNAS 2021 Vol. 118 No. 47 e2109889118, <https://doi.org/10.1073/pnas.2109889118>. How does entropy production enter the presented model?
- 2) Another missing reference is J. Pausch et al. "Time-dependent branching processes: a model of oscillating neuronal avalanches", Sci. Rep. (2020) 10:13678, <https://doi.org/10.1038/s41598-020-69705-5>. A brief contrasting comment to that paper would be helpful.
- 3) One major concern is the role of the threshold e for avalanches. The paper focuses on one choice of threshold of $2.9SD$. In the supplement Fig. S11 another threshold of $3.3SD$ is used and it's observed to produce a better fit in an example setting of $K=200$ subunits. How the thresholds are chosen (i.e. what is the procedure) and what are the effects of choosing different thresholds should be explained in detail. Different thresholds can have a huge impact on power laws (see for example Font-Clos et al: "The perils of thresholding" 2015 New J. Phys. 17 043066, doi:10.1088/1367-2630/17/4/043066). For example it would be useful to see how in the inset of Fig 3D the exponent β_I changes with the threshold e . Another example would be Fig 4B where it would also be useful to see the influence of different thresholds.

4) In the manuscript, many fits were carried out, including the values for β , c , and various exponents of power laws. It is not clear what procedures were used for these fits. I would expect to find this information in the supplement. Power law fitting is notoriously difficult, see for example Goldstein et al, "Problems with fitting to the power-law distribution", Eur. Phys. J. B 41, 255 (2004), <https://doi.org/10.1140/epjb/e2004-00316-5>. The only one I found was with regards to Fig 2F for which a least squares fit was mentioned. What function was fitted is unclear. From the picture one might guess that the fit function was an affine linear function. However, it is unclear what the null hypothesis is that lead to the reported low p-value, is it the constant function? Furthermore, the reported low R^2 value indicates that whatever model function was used, it is a poor explanation of the observed variance in the observations. Hence it is questionable whether any conclusion about the presence or absence of a correlation between α and β can be made.

5) In the manuscript, claims are made about how close the observed system is to the critical point. The adaptive Ising model is new to me and I think it would be useful to make a few statements about its physical properties first (this might be a repetition from a previous paper, but it is useful to remind readers of some basic facts): Is the critical point $\beta=1$ and $c=0$ or is there a line of critical points along $\beta=1$? If not, then there is a first-order phase transition along $\beta=1$ for values of $c \neq 0$. Closeness to a critical point is determined relative to characteristic length (and time) scales. How are these scales defined here, how does the feedback mechanism influence them? A statement that a specific parameter was found to be 0.99 needs to be put into context of these characteristic scales. To illustrate my point: I could redefine β as $\beta' = \beta^{100}$. Then, the critical point is (still) at $\beta'=1$ but the data would be matched to $\beta'=0.366$ which seems to be further away from the critical point at $\beta'=1$.

Minor points:

1) I was unable to find any typos in the manuscript, but I found one wording issue in line 233: the word 'imaginary' might be confused here as meaning purely imaginary, i.e. that the eigenvalues don't have a real part. However, I believe that the eigenvalues have a negative real part and non-zero imaginary part -- but I might be wrong. In any case, a clearer statement might be better.

Reviewer #2 (Remarks to the Author):

Referee Report:

"Statistical modeling of adaptive neural networks explains coexistence of avalanches and oscillations in resting human brain" by Lombardi et al.

The authors propose a feedback driven Ising class of neuronal network models that captures both neuronal oscillations and avalanches. The model makes direct contact with human brain resting-state activity. The model also captures the dynamics over a broad range of scales, from single sensor

oscillation to large scale avalanches. In particular I liked the fitting of the model parameters to the MEG data.

The paper is very well written and organized, the results are very original, and I have no doubt that the paper would interest readers of Nature Computational Sciences. I only have some doubts and need clarification that perhaps could improve the paper.

These are not so serious concerns, but only suggestions to the authors. To prevent delays, I need not see the new version of the manuscript before publication:

In the definition of the Adaptive Ising Model, it seems that all neurons are excitatory with $J = 1$. I think that this fact should be a bit more emphasized because other models for oscillations use inhibitory neurons.

The authors say in line 170 that, irrespective of the exact setting ($dh_i/dt = -c m$ global versus local or even $dh_i/dt = -c s_i$), the mean-field results are the same. OK, this is the meaning of a mean-field approximation. But do they have any hint about the results for the non-mean-field cases, in special the binary oscillators model with $dh_i/dt = -c s_i$? Eqs. (1) would change in this case?

The use of binary variables $s_i = \{-1, 1\}$ should be discussed a bit more because for spiking neurons usually we use $\sigma_i = \{0, 1\}$. It seems to me that in this last case the equation for h would be $dh/dt = -c(2 \langle \sigma_i \rangle - 1) = -cm$. But now, the fixed-point $m^* = 0$, $h^* = 0$ corresponds to $\langle \sigma_i \rangle = 1/2$, which is a very high neuronal activity (half of the maximum frequency). But this is a simple change of variables from s_i to σ_i . Could the authors make some observations about the consequences of this change? The binary McCulloch-Pitts-Little-Hopfield neurons s_i and the activity variable σ_i are really equivalent when we introduce the feedback mechanism?

The authors say in line 369 that “the true MEG signals are best reproduced when the adaptive Ising model is tuned close to, but slightly below its critical point”. There are several models in the literature that produce adaptive parameters slightly subcritical, perhaps the authors could give some reference here. As only an example, that need not be cited, a recent paper from Menesse et al. Chaos, Solitons & Fractals, 156: 111877 (March 2022), found that a network of stochastic integrate-and-fire neurons with a feedback mechanism over h somewhat similar to the present paper produces slightly subcriticality with stationary oscillations in the range $W \in \{0, 985; 1\}$ close to the critical value $W_c = 1$ and $h \in \{-5, 5 \cdot 10^{-4}, -4 \cdot 10^{-4}\}$ close to the critical value $h_c = 0$.

In Fig. 4, I would expect reported values for the $P(s)$ and $P(d)$ distributions, but only the $\langle s \rangle$ versus d exponent is given. Is there any reason for that? Or perhaps these numbers are in another part of the manuscript?

In the Discussion, the authors acknowledge that the crackling noise scaling exponent $\langle s \rangle$ versus d for MEG data and the model is not reproduced quantitatively. As a suggestion, a similar problem has been studied in data from animals and models, with the conclusion that the quantitative difference between exponents is due to subsampling effects (Subsampled directed-percolation models explain scaling

relations experimentally observed in the brain, TTA Carvalho et al. Frontiers in neural circuits 14: 576727).

Osame Kinouchi

Reviewer #3 (Remarks to the Author):

Key Results:

The authors propose a spin model with adaptation, capable to display both scale-free avalanches (a hallmark of criticality) and scale-specific oscillations. They present an analytical investigation and a phase diagram of the model.

The theory allows fitting the model parameters on data (MEG from humans during resting state), suggesting closeness to criticality in the brain dynamics.

Validity:

The work is very well written, and accessible to a broad range of readers. The results are exposed clearly. Theoretical derivations and analysis are well described and straightforward to follow and understand.

The literature is thorough.

Originality and significance:

The contribution and novelty of the paper are clearly discussed by the authors: existing theoretical models typically do not display both synchronized oscillations and critical regimes. However, existing models are complex, do not match microscopic and macroscopic features, and do not quantitatively match the data.

Suggestions and Questions:

I only have a few comments.

Major:

-The inference procedure is a clever choice, however, it is very specific (it fits a specific shape/class of correlation function) and it requires assumptions (to be in the resonant regime below the critical point,

see line 256). Does this limit the inference to a restricted portion of the phase diagram? Is it possible to imagine a more general inference procedure, e.g. estimating model parameters, by using a procedure similar to Boltzmann learning or pseudo-likelihood maximization? Is there a technical limitation to achieving this? If this was possible, the correlation function might be an emerging property, rather than an assumed one.

-line 389, how is the size of the subsystems N/K chosen to perform the "side-by-side" comparison with the data? Is it maybe irrelevant since you are in scale-free conditions? (i think this point is also mentioned in the discussion at line 644, but I don't understand how the choice of the extra parameters is performed).

Minor:

-line 352, the variable "n" is not introduced in the main text (it is only described in the supplementary information).

-line 239, even though it is intuitive, thanks to figure 1, maybe it would be beneficial to a general reader if the variables a_0 (zero crossing areas) and t (reversal times) were better introduced.

Author Rebuttal to Initial comments

Dr. Fabrizio Lombardi
Institute of Science and Technology Austria
Am Campus 1, 3400 Klosterneuburg, Austria
EMAIL: fabriz.lombardi@gmail.com

Ananya Rastogi, PhD
Associate Editor
Nature Computational Science

Re: Revised Nature Computational Science manuscript NATCOMPUTSCI-22-0681

Dear Dr Rastogi,

Thank you for obtaining three reviewer reports on the manuscript "*Statistical modeling of adaptive neural networks explains coexistence of avalanches and oscillations in resting human brain*".

We are pleased that the reviewers found the work interesting, relevant, and original, and that their assessment of the manuscript was very positive overall.

At the same time, the reviewers have made important comments, which helped us improve the manuscript considerably.

Following the recommendations of the Editor, to address reviewers' comments we have: (i) included in the manuscript a more detailed comparison of our model with previous works, including the work suggested by Reviewer 1; (ii) analyzed and discussed entropy production, and added a dedicated section to the SI (Section 1.1); (iii) included in the manuscript and SI further details about signal threshold selection for data-model comparison, and new Supplementary Figures that illustrate the effect of choosing different thresholds (Figures S12-S14, S19); (iv) included in the manuscript a paragraph on the choice of the subsystems size N/K , and new Supplementary Figures to better illustrate effects of different N/K on data-model comparison (Figures S9-S10); (v) added to the SI a new section on parameter fitting and compared our inference procedure with a more general method (Section 4); (vi) extended the mean field analysis of the model to discuss effects of local feedback and change of variables (SI, Section 1.2 and 1.3) (vii) included a new Figure in the SI showing the distribution of avalanche durations (Figure S20).

We thank all reviewers for their insightful comments. What follows is a point-by-point response to the comments of the reviewers.

We hope that the revised manuscript is now suitable for publication in Nature Computational Science.

Thank you very much for your time and kind consideration in this matter.

Sincerely,

Fabrizio Lombardi, Selver Pepić, Oren Shriki, Gašper Tkačik, and Daniele De Martino

Reviewer 1

Comment 1) *The presented model is a new variation of the Ising model which has been applied to Neuroscience in many publications before. Because there have been many spin-like neuronal models, the bar for promoting or establishing a new variation of these models is high. It is important to contrast previous models to the current one: what have these models accounted for and what have they missed? How does the presented model perform better compared to previous ones? Some of this already becomes clear from the manuscript but I would recommend making a structured and comprehensive comparison. One important missing reference in this context is by C. Lynn et al. "Broken detailed balance and entropy production in the human brain", PNAS 2021 Vol. 118 No. 47 e2109889118, <https://doi.org/10.1073/pnas.2109889118>. How does entropy production enter the presented model?*

Authors' reply: We thank the Reviewer for this important remark and for pointing out an interesting work that was out of our radar.

The Reviewer correctly poses our model within the broader area of Ising models as applied in neuroscience, ranging from the transformative work by Hopfield (PNAS 79, 2554-2558, 1982) to recent data-driven inverse modeling approaches (E. Schneidman et al, Nature 440.7087, 2006: 1007-1012; Tkačik et al, PNAS 2010; Roudi et al, PRE 79, 2009; Fraiman et al, PRE, 2009). Specifically, the initial formulations of the maximum-entropy problem aimed at modeling the stationary distribution from which activity patterns were drawn independently. Following work then focused on the temporal correlation structure in the neural activity by modeling across-time interactions between individual neurons (Tyrcha et al, J. Stat. Mech., 03005, 2013; Marre et al, Phys. Rev. Lett. 102, 138101, 2009; Nasser et al, J. Sta. Mech., 03006, 2013). Although these generalizations capture patterns of short-term temporal correlations and statistical criticality from data, the rhythmic behavior of brain oscillations is beyond the practical scope of these models. Our adaptive Ising model can be seen as an extension of those models that enables oscillations and permits to account for the multiple—and often contrasting—emerging collective behaviors in brain activity.

These aspects are discussed in the revised subsection "Connections to results in statistical physics" of the Discussion, line 656.

Another shortcoming of the existing models is their static character, an assumption often motivated in terms of computational tractability, that in turn affects the potential for data-driven applications. For instance, the inverse modeling of the symmetric SK model (a simplified, restricted version of the one used in the article mentioned by the Reviewer) is known to be a very difficult computational problem that is currently under intense investigation in the statistical physics community (Nguyen et al, Advances in Physics, 66, 197-261, 2017), since the easier direct problem of assessing the thermodynamics of the SK itself requires mastering a full replica

symmetry breaking scheme (Mezard, Parisi and Virasoro 1987).

We thank the referee for pointing out Lynn et al which we now cite. We note, however, that it would be difficult to assess if the model used in Lynn et al has phase transitions/bifurcations, and how they affect the dynamics. This aspect is particularly relevant to neuroscience (Izhikievich, 2007), where the focus has mainly been on the dynamical response of single neurons. We believe our work instead helps in extending this analysis to neural networks. Furthermore, in Lynn et al there is no clear assessment of how the lack of detailed balance and entropy production arise: is there an out-of-equilibrium phase transition from different regimes? How much is the lack of detailed balance a built-in hypothesis or a consequence of a dynamical bifurcation? For instance, the symmetric SK, as well as simpler kinetically constrained models, are known to undergo dynamical phase transitions that put them out-of-equilibrium spontaneously, without requiring asymmetric interactions (Ritort et al, *Advances in physics*, 52, 219-342, 2003).

In contrast, the model we use here is *both* inherently dynamical (out-of-equilibrium) and analytically tractable, showing a crispy phase diagram whose bifurcation lines/point can be *analytically* connected with the entropy production behavior, as we show below.

In the revised manuscript we discussed these aspects in relation to Lynn et al, and other relevant works in the subsection “Connections to results in statistical physics” of the Discussion. In particular, with reference to Lynn et al, we write (line 686): “Recently, an asymmetric Sherrington-Kirkpatrick (SK) model has been employed to explain broken detailed balance and entropy production in the brain as a consequence of local asymmetric interactions [8]. However, such property could also arise from an out-of-equilibrium phase transition between different regimes, as, for instance, in the symmetric SK model. In contrast, the adaptive Ising class is inherently dynamical (out-of-equilibrium) and analytically tractable, with a clear phase diagram whose bifurcation lines can be analytically connected with entropy production behavior.”

Concerning the entropy production, we added the following text and Fig. 1 to the SI (Section 1.1, Entropy production), and discussed it in the revised manuscript.

“The model has a clearly identified Hamiltonian and free energy, and the entropy production can be assessed with tools of stochastic thermodynamics [13]. We have that the instantaneous rate of “work” made by the system is ($M = mN$)

$$\dot{w} = -M \frac{dh}{dt} \quad (1)$$

[11]. Assuming that, on average, the system is stationary, the entropy rate will be (brackets stand both for averages in time and over ensembles)

$$\langle \dot{S} \rangle = - \left\langle M \frac{dh}{dt} \right\rangle. \quad (2)$$

Figure 1: Entropy production as a function of the power in the alpha band of MEG signals.

In our model this will simply be

$$\langle \dot{S} \rangle = c \langle m^2 \rangle N. \quad (3)$$

Essentially, the entropy production coincides with the “order parameter”, and becomes extensive above the critical point, i.e. for $\beta > \beta_c = 1$,

$$\langle \dot{S} \rangle = c(\beta - 1)N. \quad (4)$$

In the regime characteristic of our MEG signals, $\beta < \beta_c = 1$, the entropy production rate is sub-extensive (with a singularity at β_c , that is eventually rounded-off taking into account finite size effect), namely

$$\langle \dot{S} \rangle = \frac{c}{\beta(1-\beta)}. \quad (5)$$

For $c = 0$ the static Ising model is recovered, and, correctly, the entropy production rate becomes zero. In Fig. 1 we show the scatter plot of the inferred entropy production rate for the experimental signals versus the power in the alpha band. We observe a clear correlation between the two quantities, with the entropy production increasing linearly with the power in the alpha band (green thick line = linear fit). This suggests that an increase in entropy production in the brain could signal the approach to a dynamical bifurcation in the phase space.”

In the revised manuscript we write (line 385): “Having a well defined Hamiltonian and a free energy for our model, we can further connect the alpha band to the entropy

production (see SI, Section 1.1). For $\beta < \beta_c$, which is the regime characteristic of our MEG signals, the entropy production rate is

$$\langle \dot{S} \rangle = \frac{c}{\beta(1-\beta)}, \quad (6)$$

the inferred entropy production for the MEG signals grows linearly with the power in the alpha band (SI, Section 1.1). This suggests that an increase in entropy production in the brain could signal the approach to a dynamical bifurcation in the phase space.²

With reference to the spiraling patterns shown in Lynn *et al.*, in our framework trajectories in the phase space can be reconstructed by means of Hilbert transform (Signal Processing, 165, 115-127, 2019) (Fig. 2). These patterns clearly indicate the presence of a stable focus in phase space around which the system revolves stochastically by finite size effects, and the degree of unwinding can be connected analytically with the entropy production, and in turn with the nearby presence of a critical point.

Figure 2: Trajectories in phase space reconstructed from two MEG signals via Hilbert transform. Inferred values of $\beta = 0.98, 0.8$.

Comment 2) Another missing reference is J. Pausch *et al.* "Time-dependent branching processes: a model of oscillating neuronal avalanches", *Sci. Rep.* (2020) 10:13678, <https://doi.org/10.1038/s41598-020-69705-5>. A brief contrasting comment to that paper would be helpful.

Authors' reply: We thank the Reviewer for pointing out this work. We find the computational method based on field theory used in this paper very interesting and advanced. The same method could be applied to our model as well, in particular to highlight analytically the role of noise and finite size stochasticity beyond the linear

regime, and in turn inspire inference methods (see also The Journal of Mathematical Neuroscience, 5, 1-35, 2015).

We added a comment about this work in the revised manuscript (line 694): “In the context of branching processes, the oscillating behaviors of neuronal avalanches have recently been modeled by introducing a time-dependent oscillating extinction rate [10], and studied with perturbative field theory. While in our model oscillations are an emergent, rather than an externally-imposed property, we notice that the same method could be applied to the adaptive Ising class to highlight, for instance, the role of noise and finite size stochasticity beyond the linear regime, and, in turn, to develop novel inference methods.”

Comment 3) *One major concern is the role of the threshold e for avalanches. The paper focuses on one choice of threshold of $2.9SD$. In the supplement Fig. S11 another threshold of $3.3SD$ is used and it's observed to produce a better fit in an example setting of $K=200$ subunits. How the thresholds are chosen (i.e. what is the procedure) and what are the effects of choosing different thresholds should be explained in detail. Different thresholds can have a huge impact on power laws (see for example Font-Clos et al: "The perils of thresholding" 2015 New J. Phys. 17 043066, doi:10.1088/1367-2630/17/4/043066). For example it would be useful to see how in the inset of Fig 3D the exponent β_{α_1} changes with the threshold e . Another example would be Fig 4B where it would also be useful to see the influence of different thresholds.*

Authors' reply: We thank the Reviewer for raising this point. We agree, as we stated in the Discussion, that transparency and robustness not only with respect to model parameters but also data analysis parameters is essential; we have thus expanded our analyses as follows:

1. *Concerning the particular baseline choice of threshold e for identifying extremes and neuronal avalanches*, we remark that, for empirical data, the threshold e is typically chosen by comparing the empirical signal amplitude distribution to a best fit gaussian; e is subsequently set to values at which the amplitude distribution deviates from that best fit gaussian (see for instance Beggs& Plenz, J. Neurosci. 2003; Shriki et al, J. Neurosci. 2013). The minimum value e at which deviations become significant depends on the particular data, e.g. LFP, EEG, or MEG, and on the neurological condition (e.g. wake vs sleep; see for instance Shriki et al, 2013, Scarpetta et al, BiorXiv 2022). The data used for this study were taken from a larger dataset collected for a previous study (Shriki et al, J. Neurosci. 2013), where the e value for which the empirical distributions start to deviate from their gaussian fit was identified. In the revised manuscript we discuss this point and clearly motivate our threshold choice. We write (line 432): “We perform data and model analysis using the same threshold value $e = 2.9 SD$. The rationale for this choice is as follows. As reported in [14] and used in this work to set n_{sub} above, amplitude deviations up to $\pm 2.7SD$ closely follow a Gaussian distribution consistent with the summation of

many uncorrelated or weakly correlated signals and not indicative of individual extremal events. Thus, we need to choose $e \geq 2.7\text{SD}$ for the threshold. Higher values will reduce the number of false positives, but increase the number of false negatives. We thus picked $e = 2.9\text{SD}$ and performed extensive robustness analyses in the SI to confirm that our key results are stable in the range $2.7\text{SD} < e < 3.1\text{SD}$ (SI, Figs S13, S14, S19), which we detail result-by-result below”.

2. *Concerning the robustness of our results to different choices of threshold e .* We extensively discuss and show (new Figs S13, S14, S19) how our analysis depends on the threshold e . In particular, we show that the model reproduces empirical distributions within their respective range of variability for $2.7 \leq e \leq 3.1$ (Figs S13, S14, S19). Concerning the analysis of extreme events, we write (line, 520): We performed data and model analysis using the same threshold value $e = 2.9 \text{ SD}$, which was fixed by comparing the amplitude distribution of MEG sensor signals and model subsystem signals m_μ . For different e values, the distributions $P(A_e)$ and $P(I_e)$ follow a similar functional behavior in both data and model. The influence of thresholding on the analysis of continuous signals has been previously investigated [3]. Here, for increasing values of e , we find that (i) the probability of large (small) A_e 's tends to decrease (increase); (ii) the probability of large (small) I_e tends to increase (decrease) (Fig. S12). These effects are more pronounced for the distribution of quiescence durations $P(I_e)$, particularly in its tail. Importantly, $P(A_e)$ and $P(I_e)$, as well as the exponent β_I , show a similar dependence on e in both MEG data and model simulations, and, as a consequence, the agreement between data and model is robust to changes in the threshold e (Figs S13 and S14). As for the dependence of avalanche size distributions on e we write (line 609): As shown for $P(A_e)$ and $P(I_e)$, the distributions of avalanche sizes also moderately depends on the threshold e . This has been previously reported both in the resting human brain and in other systems [1, 12, 7]. We find that simulated avalanche size distributions show a similar dependence to the data, and are thus in agreement with empirical distributions for a range of e values (Fig. S19). Importantly, we observe that the relationship between avalanche sizes and durations is robust to changes in e , and the exponent ζ shows no significant dependence on e (Figs S12 and S19).
3. *We systematically studied how the distributions and scaling relations depend on e for model simulations in Figure S12 (SI).* As previously shown for avalanches in different systems (Beggs & Plenz, J. Neurosci. 2003; Lombardi et al, Neurocomputing 464, 2021), for increasing values of e the likelihood of large avalanches decreases, while the likelihood for small avalanches increases, which leads to an increase of the power-law exponent. In Figure S12 (SI), we show that numerical distributions of instantaneous network activations, $P(A_e)$, and avalanche sizes, $P(s)$, exhibit a similar dependence on e .

Comment 4) *In the manuscript, many fits were carried out, including the values for beta, c, and various exponents of power laws. It is not clear what procedures*

were used for these fits. I would expect to find this information in the supplement. Power law fitting is notoriously difficult, see for example Goldstein et al, "Problems with fitting to the power-law distribution", *Eur. Phys. J. B* 41, 255 (2004), <https://doi.org/10.1140/epjb/e2004-00316-5>. The only one I found was with regards to Fig 2F for which a least squares fit was mentioned. What function was fitted is unclear. From the picture one might guess that the fit function was an affine linear function. However, it is unclear what the null hypothesis is that lead to the reported low p-value, is it the constant function? Furthermore, the reported low R^2 value indicates that whatever model function was used, it is a poor explanation of the observed variance in the observations. Hence it is questionable whether any conclusion about the presence or absence of a correlation between alpha and beta can be made.

Authors' reply: We thank the Reviewer for pointing out that details on data fitting procedures were missing, which we have corrected in the revised manuscript as described below.

1. *Fitting the parameters (β, c) of the model.* We now write in revised SI, Section 4:

The parameters β and c were chosen as the minimizers of L_2 error $D(c, \beta)$ between the discrete autocorrelation function estimated from data and the approximate analytical expression of the latter in the model, i.e.

$$D(c, \beta) = \frac{1}{T} \sum_{t=1}^T (C_{exp}(t) - C_{mod,\beta,c}(t))^2 \quad (7)$$

where

$$C_{mod,\beta,c}(t) = e^{-\gamma t} \left(\cos(\omega t) + \frac{\gamma \sin(\omega t)}{\omega} \right) \quad \beta = 1 - \gamma \quad c = \frac{\gamma^2 - \omega^2}{4(1 - \gamma)} \quad (8)$$

Autocorrelation function is estimated from the sensor data (z-scored by its mean and variance $x(t) \rightarrow \frac{x(t) - \bar{x}}{\sigma}$ to get a zero-mean unit variance signal):

$$C_{exp}(\tau) = \frac{1}{N - \tau} \sum_{t=0}^{N-\tau} x_t x_{t+\tau}, \quad (9)$$

which is a standard (maximum-likelihood) estimator for the autocorrelation assuming IID gaussian noise. We typically use $N = O(10^5)$ samples, sufficient for this procedure to converge to estimates with very small empirical error; our standard error for $C(\tau)$ is of the order 10^{-3} , estimated via bootstrap. The function $D(\beta, c)$ is an analytic function of two variables whose minima can be found with elementary methods, in our case, gradient descent.

2. *Fitting size and duration power-law distributions.* Fitting power laws to distributions, especially if undersampled in the tails, can be statistically challenging,

as the Reviewer points out. We used such fitting only in Fig. 1 on samples simulated from the adaptive Ising model, as we can draw essentially unlimited number of samples from our simulation, making the fits robust. Originally, we had performed a ordinary linear least square (LS) fit on logarithmically transformed data, i.e. $\log(x)$ vs $\log P(x)$. In the revised manuscript, we report the maximum likelihood estimates (MLE) for the power law exponents in Fig. 1, and state in the caption that “power-law fits were performed using a maximum likelihood estimator [31]”. We note that the two estimators give consistent results as we are not sample limited in this simulated scenario: specifically, the MLE are $\tau = 1.227 \pm 0.004$ for $a_0 \in [0.1, 100]$; $\alpha = 1.3617 \pm 0.002$ for $t \in [2, 500]$ (LS estimates were $\tau = 1.29 \pm 0.01$; $\alpha = 1.39 \pm 0.01$). In all subsequent figures where we do data/model comparisons, however, we did not use power-law fitting anywhere, but directly estimated Kullback-Leibler divergence between complete (data and model) distributions. This measure of distribution (dis)similarity does not assume any *a priori* functional form for the distributions.

3. *Fitting scaling relationships.* Importantly, scaling relationships in Fig 3D and Fig 4 insets are not power-law fits to distributions or distributional data, but rather standard linear regressions between a statistic and a control parameter. Here, a standard linear least square fit on logarithmically transformed dependent and independent variables is appropriate. We apologize for not providing this technical information, which we have remedied in the revised manuscript by writing: (caption of Fig. 3) “The exponent β_f was estimated via ordinary linear least square fit $y = ax + b$, with $y = \ln(-\ln P_b)$ and $x = \ln(\epsilon)$.”; (caption of Fig. 4) “The exponent ζ was estimated via ordinary linear least square fit $y = ax + b$, with $y = \ln(s)$ and $x = \ln(\epsilon^{-3/2}d)$ ”.
4. *Linear correlations in Figs. 2D and 2F.* We fitted a linear model $y = ax + b$, and the reported *p*-value refers to the non-zero linear regression coefficient *a* (specifically, the null hypothesis that is being rejected at the reported *p*-value significance is $y = \text{const}$). We added this information in the caption of Fig. 2 and in Section 5 of the Methods, where we write: “The reported *p*-value for the relationship between β and the power in the alpha band in Figure 2D and β and α in Figure 2F rejects the null hypothesis that $y = \text{const}$ (in other words, it is the significance of the nonzero linear regression coefficient).” In case of Fig 2F (but not 2D) we agree with the Reviewer that the R^2 is not very high, i.e., the α does not explain a lot of variance in β , but this is a claim we did not wish to make or imply. α may be influenced by many other factors and can be quite strongly affected by the variability across the subjects; while small, the correlation is, however, statistically very significant.

Comment 5) *In the manuscript, claims are made about how close the observed system is to the critical point. The adaptive Ising model is new to me and I think it would be useful to make a few statements about its physical properties first (this*

might be a repetition from a previous paper, but it is useful to remind readers of some basic facts): Is the critical point $\beta=1$ and $c=0$ or is there a line of critical points along $\beta=1$? If not, then there is a first-order phase transition along $\beta=1$ for values of $c \neq 0$. Closeness to a critical point is determined relative to characteristic length (and time) scales. How are these scales defined here, how does the feedback mechanism influence them? A statement that a specific parameter was found to be 0.99 needs to be put into context of these characteristic scales. To illustrate my point: I could redefine β as $\beta' = \beta^{100}$. Then, the critical point is (still) at $\beta'=1$ but the data would be matched to $\beta'=0.366$ which seems to be further away from the critical point at $\beta'=1$.

Authors' reply: We thank the Reviewer for raising this very important points, which we addressed in the revised manuscript and SI. Specifically:

1. *Concerning the critical/bifurcation point*, we remark that our system in the thermodynamic limit—where phase transitions are defined (Ruelle 1999)—is described by a system of ordinary differential equations that can be analyzed by means of bifurcation theory. Because it is confined, our system verifies the hypothesis of the Hopf theorem (Hopf, 1942), and from linear perturbation analysis there is a pair of complex eigenvalues becoming purely imaginary at $\beta_c = 1$. Thus, this is a line of Andronov-Hopf bifurcations where a focus loses stability in favour of an emerging limit cycle. The first/second order classification refers to singularities in the derivative of free energy in equilibrium static transitions, and it does not usually apply in the context of out-of-equilibrium systems where a variational principle is lacking. In the case of Andronov-Hopf bifurcations there is an analogous classification of continuous/discontinuous emergence that is called supercritical and sub-critical, respectively (Izhikievich 2007). These two cases can be differentiated by calculating the sign of the so-called Lyapunov coefficient λ_1 . We calculated it analytically for our case (see Izhikievich 2007 for details) and found

$$\lambda_1 = -(1+c)\beta/8 < 0 \quad (10)$$

meaning that we are in the presence of a supercritical (continuous) bifurcation. To clarify this point in the revised manuscript we write (line 238): “In the resonant regime, $c > c^*$, oscillations become more prominent as the critical point $\beta_c = 1$ is approached, finally transitioning into self-sustained oscillations for $\beta > \beta_c$ (Fig. S2). At $\beta = \beta_c$ we have a line of Andronov-Hopf bifurcations where a focus loses stability and a limit cycle emerge. We find that this bifurcation is supercritical, the first Lyapunov coefficient being negative, i.e. $\lambda_1 = -(1+c)\beta/8 < 0$ [6]”

2. *As for the redefinition of β* , this is the Lagrange multiplier of the energy in our model and the set of admissible transformations on β is usually restricted

to conformal ones, the most meaningful being in particular a change of unit $\beta \rightarrow a\beta'$. For this case, performing the inference one would retrieve the same value divided by a . We would also like to stress that we set $J = 1$ and $\beta_c = 1$, and thus what we infer in essence is

$$\frac{\beta J}{(\beta J)_c}. \quad (11)$$

It is possible to connect β with the typical timescales of the system analytically through the result of the linear stability analysis. The relaxation/decorrelation time of the system is in fact

$$\gamma = \frac{2}{1 - \beta}. \quad (12)$$

We fully acknowledge that in general the issue raised by the Reviewer is a very interesting one and there is a very promising line of research about defining “inherent” metrics in phase diagrams able to take into account possible inference biases (Janke et al, Physica A 2004, 336: 181-186). In essence given a stochastic trajectory $m(t)$, if one has the expression of the probability conditioned on the parameters (i.e., a likelihood function), $P(m(t)|\vec{u} = (\beta, c))$, this metric is given by the so-called Fisher information matrix

$$g_{ij} = \left\langle \frac{\partial^2 \log P}{\partial u_i \partial u_j} \right\rangle \quad (13)$$

where the average is performed over stochastic trajectories. Such evaluation could potentially be performed with the field theory tools used in the reference suggested by the Reviewer (Sci. Rep. (2020) 10:13678).

We have included these considerations into the SI, Section 4. Furthermore, we highlighted in the revised manuscript the definition of relaxation time. We write (line 266): “In the resonant regime below the critical point ($c > c^*$, $\beta < \beta_c$), it is possible to analytically compute the autocorrelation function of the ongoing network activity $m(t)$ in the linear approximation [5]:

$$C(\tau) = e^{-\gamma\tau} \left(\cos \omega\tau + \frac{\gamma}{\omega} \sin \omega\tau \right), \quad (14)$$

where $\gamma = (1 - \beta)/2$ is the relaxation time of the system, and $\omega = \sqrt{\beta c - (1 - \beta)^2/4}$ is the characteristic angular frequency of the model.”

Comment 6) *I was unable to find any typos in the manuscript, but I found one wording issue in line 233: the word ‘imaginary’ might be confused here as meaning purely imaginary, i.e. that the eigenvalues don’t have a real part. However, I believe that the eigenvalues have a negative real part and non-zero imaginary part – but I might be wrong. In any case, a clearer statement might be better.*

Authors’ reply: Thank you for pointing this out: in the revised manuscript we changed “imaginary values” in “complex values”.

Reviewer 2

Comment 1) *In the definition of the Adaptive Ising Model, it seems that all neurons are excitatory with $J = 1$. I think that this fact should be a bit more emphasized because other models for oscillations use inhibitory neurons.*

Authors' reply: We thank the Reviewer for this remark. As correctly pointed out by the Reviewer, all neurons that are explicitly simulated as Ising-like spins in the model are excitatory.

We explicitly state that in the revised manuscript, and write (line 146): **In the simplest, fully homogeneous scenario described here, neurons interact with each other through synapses of equal strength $J_{ij} = J = 1$, namely they are all excitatory.** However, we note that, as we state in the manuscript (line 158), “negative feedback can be identified with a mean-field approximation to the inhibitory neuron population that uniformly affects all excitatory neurons with a delay given by the characteristic time c^{-1} ”. In other words, our model can be interpreted as having effective inhibition mediated by the negative feedback loop.

The model is easily extensible to inhibitory neurons that can be individually simulated as spins as well. For instance, one could define an arbitrary vector ξ of ± 1 values and consider the hamiltonian

$$H = -\frac{1}{N} \sum_{i < j} s_i \xi_i \xi_j s_j \quad (15)$$

and as well as add a spatially heterogeneous external field term h along the ξ s i.e.

$$H = -\frac{1}{N} \sum_{i < j} s_i \xi_i \xi_j s_j - h \sum_i \xi_i s_i \quad (16)$$

in this case neuronal interactions can be of any kind $J_{ij} = \xi_i \xi_j$ (excitatory as well as inhibitory), but upon performing the gauge transformation $S_i = s_i \xi_i$ one discovers that this is equivalent to the simple Ising hamiltonian

$$H = -\frac{1}{N} \sum_{i < j} S_i S_j - h \sum_i S_i. \quad (17)$$

This calculation suggests that, as far as one is concerned with symmetric hamiltonian models, the basic dynamics is not affected by changes in the character of the connections (excitatory, as in our simulations, or excitatory and inhibitory). On the other hand, different dynamical behaviors could be obtained by increasing the complexity of the free energy landscape (for instance adding frustration).

With regard to asymmetric models, in Methods, Section 1 we have a subsection, “Mapping between the adaptive Ising model and an E-I network”, in which we show that in the linear regime a mapping can be established between our fully excitatory feedback model with more classical excitatory-inhibitory (E-I) two-population model, which in a deterministic setting is a standard model for producing oscillations.

Comment 2) *The authors say in line 170 that, irrespective of the exact setting (global versus local[...]), the mean-field results are the same. OK, this is the meaning of a mean-field approximation. But do they have any hint about the results for the non-mean-field cases, in special the binary oscillators model with $dh_i/dt = -cs_i$? Eqs. (1) would change in this case?*

Authors' reply: This is a very interesting question that is technically difficult to assess in general if one wants to keep analytical control.

To make an attempt at this point, we added the following mean field calculation for the adaptive Ising model to the revised SI (Section 1.2).

In the simple mean field setting that we consider for our adaptive Ising model, the extension from global to local fields shall not affect the overall dynamical behavior with regard to average quantities, as we show in the following.

Consider a fully connected model with local fields, h_i , whose Hamiltonian is

$$H = -\frac{J}{N} \sum_{i < j} s_i s_j - \sum_i h_i s_i \quad (18)$$

The partition function can be calculated by neglecting second order terms (Curie-Weiss approximation, where in the quadratic term in the Hamiltonian above we approximate $s_j \sim \langle s \rangle$):

$$Z = \sum_{\mathbf{s}} e^{-\beta H} \sim \sum_{\mathbf{s}} \prod_i e^{\beta(J\langle s \rangle + h_i)s_i} = \prod_i 2 \cosh(\beta(J\langle s \rangle + h_i)) \quad (19)$$

By including the feedback and applying linear response to $m_i = \langle s_i \rangle$, one has the system of dynamical differential equations (using $m = \langle s \rangle = \frac{1}{N} \sum_i \langle s_i \rangle$ and $J = 1$):

$$\begin{aligned} \dot{m}_i &= -m_i + \tanh(\beta(m + h_i)) \\ \dot{h}_i &= -cm_i \end{aligned} \quad (20)$$

If one sets $m_i = m$ for each i , this system has the same solution(s) as the homogeneous system under the influence of a global field. In what follows, we indicate the global magnetization and field as $m_g(t)$ and $h_g(t)$, respectively.

We can test the stability of these solutions by considering m_i and h_i as small perturbations to the global fields, namely $m_i = m_g + \delta_i$, $h_i = h_g + \delta_{hi}$. By substituting such expressions into the Eqs (21) and using $\tanh(a + b) = (\tanh(a) + \tanh(b))/(1 + \tanh(a) \cdot \tanh(b))$ and neglecting higher order terms, we obtain the linearized system

$$\begin{aligned} \dot{\delta}_i &= -\delta_i + \beta \delta_{hi} (1 - \beta \tanh^2(\beta(m_g + h_g))) \\ \dot{\delta}_{hi} &= -c \delta_i \end{aligned} \quad (21)$$

If $\beta < \beta_c$, $m_g = h_g = 0$, the eigenvalues of the Jacobian

$$\lambda_{\pm} = \frac{-1 \pm \sqrt{1 - 4\beta c}}{2} \quad (22)$$

Figure 3: Magnetization as a function of time from numerical simulations of the system of ODE's (21) for three different spins and for $c = 0.01$, $\beta = 0.95$ and $N = 100$ variables. The global magnetization, m_g , is shown in purple. Initial conditions are random in $(-1, 1)$. The initial perturbations applied to the different spins all eventually decay, and all m_i converge to the same value $m = \langle m \rangle = 0$.

have a negative real part, i.e., perturbations get exponentially suppressed. This is confirmed by numerical simulations of the system of Eqs (21) as illustrated in Fig 3.

Taken together, this mean field approximation suggests that the system with local feedback retains qualitatively the same features as for the global feedback case. A full assessment of the impact of local fields, beyond simple mean field approximations, would require further in-depth investigations.

Comment 3) *The use of binary variables $s_i = -1, 1$ should be discussed a bit more because for spiking neurons usually we use $\sigma_i = 0, 1$. It seems to me that in this last case the equation for h would be $dh/dt = -c(2 \langle \sigma_i \rangle - 1) = -cm$. But now, the fixed-point $m^* = 0$, $h^* = 0$ corresponds to $\sigma_i = 1/2$, which is a very high neuronal activity (half of the maximum frequency). But this is a simple change of variables from s_i to σ_i . Could the authors make some observations about the consequences of this change? The binary McCulloch-Pitts-Little-Hopfield neurons s_i and the activity variable σ_i are really equivalent when we introduce the feedback mechanism?*

Authors' reply: The Reviewer is correct. To deal with such a change of variables and the associate shift in the stationary neural activity, we can modify the model to take into account the level of activity by inserting a constant feedback that fixes it to some desired value $m_0 \neq 0$ ($m_0 = -0.99$ would correspond in terms of the σ variables to very low activity). In the following we show that, in the mean field approximation, the model with an $m_0 \neq 0$ is qualitatively equivalent to the model

with no bias ($m_0 = 0$). The main difference in this case is that the bifurcation line is located at $\beta_c = 1/(1 - m_0^2)$. We have added this discussion to the revised SI (Section, 1.3).

The ODE system for the model with a bias $m_0 \neq 0$ on the magnetization is

$$\dot{m} = -m + \tanh(\beta(m + h)) \quad (23)$$

$$\dot{h} = -c(m - m_0) \quad (24)$$

The stationary point will be then

$$m_s = m_0 \quad (25)$$

$$h_s = h_0 = \frac{\text{atanh}(m_0)}{\beta} - m_0 \quad (26)$$

A linear perturbation analysis shows that the critical point will be now at

$$\beta_c = \frac{1}{1 - m_0^2}. \quad (27)$$

This is once again an Andronov-Hopf bifurcation line where the real part of a pair of complex eigenvalues changes sign, indicating the emergence of a limit cycle. The bifurcation line is preceded by a resonant regime as in the case for $m_0 = 0$. Beyond the critical point, an approximate analytical solution can be worked out for $\beta \gtrsim \beta_c$ (harmonic oscillations) by a two-time expansion (Strogatz 2018). If we set $\epsilon = \beta - \beta_c$ we have

$$m - m_0 \sim \sqrt{\epsilon} \cos\left(\left(1 + \frac{1}{2}\epsilon\right)\sqrt{c}t + \phi_0\right), \quad (28)$$

which gives an oscillating magnetization as in the case $m_0 = 0$. This shows that our framework can accommodate for a shift in global activity levels—or a reparametrization of spin variables s_i from $(-1, +1)$ to $(0, 1)$ —and the qualitative behavior of the model is preserved, at least in the simple mean field setting.

Comment 4) *The authors say in line 369 that “the true MEG signals are best reproduced when the adaptive Ising model is tuned close to, but slightly below its critical point”. There are several models in the literature that produce adaptive parameters slightly subcritical, perhaps the authors could give some reference here. As only an example, that need not be cited, a recent paper from Menesse et al. Chaos, Solitons & Fractals, 156: 111877 (March 2022), found that a network of stochastic integrate-and-fire neurons with a feedback mechanism over h somewhat similar to the present paper produces slightly subcriticality with stationary oscillations in the range $W \in (0, 985; 1)$ close to the critical value $W_c = 1$ and $h \in (-5, 5 * 10^{-4}, -4 * 10^{-4})$ close to the critical value $h_c = 0$.*

Authors’ reply: We thank the Reviewer for suggesting us this interesting article. In the revised manuscript we included some references to this and other models that show slightly subcritical states. We write (line 728): “Other models also predict adaptive parameters that are slightly sub-critical [9].”

Comment 5) *In Fig. 4, I would expect reported values for the $P(s)$ and $P(d)$ distributions, but only the versus d exponent is given. Is there any reason for that? Or perhaps these numbers are in another part of the manuscript?*

Authors' reply: The distribution of avalanche durations, $P(d)$, is now shown in the revised SI (Fig. S20).

Comment 6) *In the Discussion, the authors acknowledge that the crackling noise scaling exponent versus d for MEG data and the model is not reproduced quantitatively. As a suggestion, a similar problem has been studied in data from animals and models, with the conclusion that the quantitative difference between exponents is due to subsampling effects (Subsampled directed-percolation models explain scaling relations experimentally observed in the brain, TTA Carvalho et al. *Frontiers in neural circuits* 14: 576727).*

Authors' reply: Thank you, this is a very relevant reference, which we now include in the revised manuscript. We write (line 769): “Regarding this point, recent numerical work showed that one can obtain $\zeta \simeq 1.3$ by subsampling the activity of models that are otherwise constructed to have $\zeta = 2$ [2], which suggests that subsampling in brain activity recordings could alternatively explain $\zeta \simeq 1.3$, the value reported here and in other studies [4].”.

Reviewer 3

Comment 1) *The inference procedure is a clever choice, however, it is very specific (it fits a specific shape/class of correlation function) and it requires assumptions (to be in the resonant regime below the critical point, see line 256). Does this limit the inference to a restricted portion of the phase diagram? Is it possible to imagine a more general inference procedure, e.g. estimating model parameters, by using a procedure similar to Boltzmann learning or pseudo-likelihood maximization? Is there a technical limitation to achieving this? If this was possible, the correlation function might be an emerging property, rather than an assumed one.*

Authors' reply: We thank the Reviewer for this remark. In the revised SI we compared our parameter fitting procedure with an alternative inference method (SI, Section 4.1), as discussed in the following. We referred to it in the revised manuscript, line 276 and 728.

Our inference procedure is specific for the range $\beta < \beta_c$. In our case, this is fully justified by the exponential decay that characterizes the autocorrelation of all MEG sensor signals. A general inference procedure, not restricted to one half of the phase diagram and going beyond our linear approximation, would require estimating the Fisher Information Matrix to perform inverse modeling (Janke et al, *Physica A*, 336 (1–2), 181–186, 2004)).

For the case of second order stochastic differential equations, several inference methods have been recently proposed (Ferretti et al, *Physical Review X* 10, 031018, 2020;

Figure 4: Compute time versus the number data points for the Ferretti method (FM), (Physical Review X 10, 031018, 2020.)

Brückner et al, Physical review letters 125, 058103, 2020). These methods can in principle be used on Eqs (1) from the main paper directly.

To address the point raised by the Reviewer, we have implemented the inference method from Ferretti et al, and compared it with our approach to parameter fit. First, we notice that, although Ferretti method (FM) time complexity is polynomial in the length of the time series N , FM is considerably more computationally demanding as it scales approximately as N^3 (Fig. 4)—in contrast to our autocorrelation matching that is linear in N . For our setting, this is essential, as our timeseries have $N \sim 10^5$ time points; the time complexity of FM becomes such that we could only perform the comparison between the Ferretti method (FM) and our autocorrelation matching (AM) using multiple chunks of $N = 1000$ data points each.

We first tested FM on time series generated by simulating our model at different values of β and c . We found that the parameters inferred via FM are in excellent agreement with those obtained via AM, and with the underlying ground truth (Fig. 5). We note that, for the inference with FM, we had to use the time series $h(t)$, i.e. the integral of the magnetization $m(t)$, which is much smoother than $m(t)$. Indeed, for $m(t)$ we found that the FM algorithm does not converge, suggesting that the FM is sensitive to the amount of noise present in the time series.

Given the fragility of the FM algorithm with respect to noise, we wondered whether it can estimate parameters well from empirical data. We compared the FM inference from $h(t)$ of a synthetic signal with the $h(t)$ of a real signal with the same length and exactly matching autocorrelation function (and thus identical β as identified by AM). Indeed, FM produced significantly different estimates of β for the real and synthetic signals. To verify that these issues are not due to the limited

Figure 5: Inferred β (top) and c (bottom) by the Ferretti method (FM; y axis) and the autocorrelation matching (AM; x axis) on synthetic data generated from model simulations. The cyan tick line represents equality ($y = x$). AM recovers the ground truth by construction. We considered time traces simulated from the model in four different conditions $\beta = 0.8, 0.85, 0.9, 0.98$ and $c = 0.01, 0.02, 0.03, 0.04$, respectively.

number of data points used to perform the inference with FM, we further tested FM on synthetic signals with different noise levels obtained from a linear combination between $h(t)$ and $m(t)$, namely $s(t) = h(t) + bm(t)$ with $0 < b < 1$. By construction, such signals have a fixed β , c , and an autocorrelation that is invariant of b . We found that, already for b as small as 0.1, the FM underestimates β (Fig. 6); the bias of the FM method thus depends on b . For larger b values, the method did not converge. This suggests that noise plays a key role in the FM performance. In contrast, AM identifies the parameters robustly (by construction) for any value of b . Despite our attempts, on our data, FM can thus not provide a reliable benchmark against which

to compare and calibrate the performance of our autocorrelation-matching (AM) approach.

Figure 6: Inferred β from the Ferretti method (FM) across chunks of a synthetic time series $s(t) = h(t) + 0.1m(t)$ obtained combining h and m from a model simulation with $\beta = 0.97$. Each chunk contains 1000 data points. The cyan tick line represents the ground truth. Already a small addition of $m(t)$ biases the inference of β downward from the ground truth value with the FM method.

Lastly, we compared FM and AM on empirical MEG signals in greater detail. For the parameter c , we found a substantial agreement between the two methods, but the inferred errors of the FM were large due to the limit on timeseries length. As for β , we found that both methods consistently place the system slightly below the critical point $\beta_c = 1$ (Fig. 7), but with significant biases likely due to the application of the FM method to noisy empirical data, as explained above.

Overall these results indicate that, for controlled signals with low noise, FM and AM provide results that are in quantitative agreement with each other and with the ground truth (Fig. 5). This shows that our parameter estimates via autocorrelation are consistent with an inference method that does not assume a specific form of the autocorrelation, when that method is applicable.

Although not exhaustive, our analysis further indicates that observed biases in FM do not come from using a limited number of data points ($N = 1000$), but rather from the noise present in the real signals. Indeed, the FM shows a general instability even on synthetic data if the noise is too large. This plus other non-linear/non-gaussian properties of the MEG data may be responsible for the discrepancy observed in real data between the two methods. These difficulties, as well as the computational complexity of the FM, lead us to conclude that AM (subject to more a priori restrictions on the signal class) is nevertheless preferable for our

Figure 7: Inferred β (top) and c (down) from the Ferretti method (FM; y axis) and the autocorrelation matching (AM; x axis) for 10 MEG signals in the α band.

application to the FM (which is theoretically more powerful and generic, but seems to suffer from technical limitations).

Finally, we note that we implemented the Ferretti algorithm to fit a linear model, as shown in Ferretti et al, Phys. Rev. X, 10, 031018, 2020. To probe the dynamics beyond linear terms, ongoing work is focused on the recent inference method developed by Brückner et al, Phys. Rev. Lett. 125, 058103, 2020.

Comment 2) line 389, how is the size of the subsystems N/K chosen to perform the "side-by-side" comparison with the data? Is it maybe irrelevant since you are in scale-free conditions? (i think this point is also mentioned in the discussion at line 644, but I don't understand how the choice of the extra parameters is performed).

Authors' reply: We thank the Reviewer for pointing out that this information

was missing. In the revised manuscript we added a paragraph to explain how the the number of subsystem size, $n_{sub} = N/K$, was selected to perform the data-model comparison. We write (line 421): “The number of neurons n_{sub} in each subsystem is fixed by matching the amplitude distribution of m_{μ} to the MEG sensor amplitude distribution between $\pm 2.7SD$, the range over which amplitude distributions follow a Gaussian behavior [14]. This procedure gives the sufficient number of neurons whose collective activity accounts for the the Gaussian core of the empirical signal amplitude distribution, thus providing a common reference to consistently define the extreme events in empirical data and model simulations.”

Furthermore, we added a new figure in the SI (Fig. S10) to better illustrate the robustness of the model-data comparison with respect to n_{sub} .

Comment 3) *line 352, the variable “n” is not introduced in the main text (it is only described in the supplementary information).*

Authors’ reply: The variable “n” is now defined in the main text (line 367). We write: “In brief, the integrated signal is divided into windows of equal length n , and the local trend is subtracted in each window.”

Comment 4) *line 239, even though it is intuitive, thanks to figure 1, maybe it would be beneficial to a general reader if the variables a_0 (zero crossing areas) and t (reversal times) were better introduced.*

Authors’ reply: Thank you. We introduced these two quantities in the main text of the revised manuscript (line 246). We write: “The reversal time, t , is defined as the time interval between two consecutive points in time at which a given signal crosses zero. Correspondingly, the zero-crossing area, a_0 , is the area under the signal curve between two zero crossing points.”

References

- [1] J. M. Beggs and D. Plenz. Neuronal avalanches in neocortical circuits. *J. Neurosci.*, 23:11167–11177, 2003.
- [2] T. T. A. Carvalho, A. J. Fontenele, M. Girardi-Schappo, T. Feliciano, L. A. A. Aguiar, T. P. L. Silva, N. A. P. de Vasconcelos, P. V. Carelli, and M. Copelli. Subsampled directed-percolation models explain scaling relations experimentally observed in the brain. *Front. Neural Circuits*, 14:576727, 2021.
- [3] F. Font-Clos, G. Pruessner, N. R. Moloney, and A. Deluca. The perils of thresholding. 17:043066, 2015.
- [4] A. J. Fontenele, N. A. P. de Vasconcelos, T. Feliciano, L. A. A. Aguiar, C. Soares-Cunha, B. Coimbra, L. D. Porta, S. Ribeiro, A. J. Rodrigues, N. Sousa, P. V.

- Carelli, and M. Copelli. Criticality between cortical states. *Phys. Rev. Lett.*, 122(20):208101, 2019.
- [5] C. Gardiner. *Stochastic Methods*, volume 4. Springer, Berlin, 2009.
- [6] E. M. Izhikevich. *Dynamical systems in neuroscience*. MIT press, 2007.
- [7] F. Lombardi, O. Shriki, H. J. Herrmann, and L. de Arcangelis. Long-range temporal correlations in the broadband resting state activity of the human brain revealed by neuronal avalanches. *Neurocomputing*, 461:657–666, 2021.
- [8] C. W. Lynn, E. J. Cornblath, L. Papadopoulos, M. A. Bertolero, and D. S. Bassett. Broken detailed balance and entropy production in the human brain. *Proc Natl Acad Sci USA*, 118(47):1–7, 2021.
- [9] G. Menesse, B. Marin, M. Girardi-Schappo, and O. Kinouchi. Homeostatic criticality in neural networks. *Chaos, Solitons & Fractals*, 156:111877, 2022.
- [10] J. Pausch, R. Garcia-Millan, and G. Pruessner. Time-dependent branching processes: a model of oscillating neuronal avalanches. *Sci. Rep.*, 10:13678, 2020.
- [11] L. Peliti. On the work-hamiltonian connection in manipulated systems. *J. Stat. Mech.*, P05002, 2008.
- [12] T. Petermann, T. C. Thiagarajan, M. Lebedev, M. Nicolelis, D. R. Chialvo, and D. Plenz. Spontaneous cortical activity in awake monkeys composed of neuronal avalanches. *Proc Natl Acad Sci USA*, 106(37):15921–15926, 2009.
- [13] U. Seifert. Stochastic thermodynamics, fluctuation theorems and molecular machines. *Rep. Prog. Phys.*, 75:126001, 2012.
- [14] O. Shriki, J. Alstott, F. Carver, T. Holroyd, R. N. A. Hanson, M. L. Smith, R. Coppola, E. Bullmore, and D. Plenz. Neuronal avalanches in the resting meg of the human brain. *J. Neurosci.*, 33(16):7079–7090, 2013.

Decision Letter, first revision:

Dear Dr. Lombardi,

Thank you for submitting your revised manuscript "Statistical modeling of adaptive neural networks explains coexistence of avalanches and oscillations in resting human brain" (NATCOMPUTSCI-22-0681A). It has now been seen by the original referees and their comments are below. The reviewers find that the paper has improved in revision, and therefore we'll be happy in principle to publish it in Nature Computational Science, pending minor revisions to satisfy the referees' final requests and to comply with our editorial and formatting guidelines.

Thank you again for your interest in Nature Computational Science Please do not hesitate to contact me if you have any questions.

Sincerely,

Ananya Rastogi, PhD
Associate Editor
Nature Computational Science

ORCID

Reviewer #1 (Remarks to the Author):

The authors have addressed all of my major concerns to my satisfaction. I believe that there remain many interesting follow-up research questions and I hope that the authors continue in this area in their future work.

Reviewer #2 (Remarks to the Author):

The authors replied to all my concerns and suggestions, and also from the other referees. I am satisfied with the changes in the manuscript.

Osame Kinouchi

Reviewer #3 (Remarks to the Author):

All my comments has been addressed, and the paper has been significantly improved.

Reviewer #3 (Remarks on code availability):

The repository provides the code (c++ and in matlab), data, and a README.

The README is well written and detailed, however the instructions on how to install and run are not present. I would suggest including the lines of code to compile the c++ source code and to run it. This would benefit non-expert users.

E.g. I have not been using c++ since many years. I tried to compile by the line:

```
>> gcc adaptive_ising.cpp
```

but I encountered an error, and I could not go further.

Also, the version of the software necessary to run code should be indicated.

Author Rebuttal, first revision:

Dear Dr Rastogi,

Attached please find the final materials for the manuscript “Statistical modeling of adaptive neural networks explains coexistence of avalanches and oscillations in resting human brain”.

We made all requested changes. In particular, we shortened the abstract (about 150 words) and the main text of the manuscript, which is now about 4000 words. Moreover, as requested by Reviewer #3, we updated the compilation instructions for the main code on the GitHub repository.

Concerning the brief summary, we propose the following text: “The study shows that scale specific oscillations and scale-free neuronal avalanches in resting brains co-exist in the simplest model of an

adaptive neural network close to a non-equilibrium critical point at the onset of self-sustained oscillations”.

We would like you to include the twitter handle @ISTAustria in the tweet that will follow the publication of our paper. We suggest the following hashtags: #neuronalavalanches, #criticality, #neuraloscillations, #brainrhythms, #neuralnetworks.

We would like to make the reviewer reports, author rebuttal letters and editorial decision letters public.

Thank you very much for your time and kind consideration. Sincerely,
Fabrizio Lombardi, Selver Pepić, Oren Shriki, Gašper Tkačik, and Daniele De Martino

Final Decision Letter:

Dear Dr Lombardi,

We are pleased to inform you that your Article "Statistical modeling of adaptive neural networks explains coexistence of avalanches and oscillations in resting human brain" has now been accepted for publication in Nature Computational Science.

Once your manuscript is typeset, you will receive an email with a link to choose the appropriate publishing options for your paper and our Author Services team will be in touch regarding any additional information that may be required.

Please note that *Nature Computational Science* is a Transformative Journal (TJ). Authors may publish their research with us through the traditional subscription access route or make their paper immediately open access through payment of an article-processing charge (APC). Authors will not be required to make a final decision about access to their article until it has been accepted. [Find out more about Transformative Journals](https://www.springernature.com/gp/open-research/transformative-journals)

Authors may need to take specific actions to achieve [compliance with funder and institutional open access mandates](https://www.springernature.com/gp/open-research/funding/policy-compliance-faqs). If your research is supported by a funder that requires immediate open access (e.g. according to [Plan S principles](https://www.springernature.com/gp/open-research/plan-s-compliance)) then you should select the gold OA route, and we will direct you to the compliant route where possible. For authors selecting the subscription publication route, the journal's standard licensing terms will need to be accepted, including [self-archiving policies](https://www.springernature.com/gp/open-research/policies/journal-policies). Those licensing terms will supersede any other terms that the author or any third party may assert apply to any version of the manuscript.

If you have any questions about our publishing options, costs, Open Access requirements, or our legal

forms, please contact ASJournals@springernature.com

Acceptance of your manuscript is conditional on all authors' agreement with our publication policies (see <https://www.nature.com/natcomputsci/for-authors>). In particular your manuscript must not be published elsewhere and there must be no announcement of the work to any media outlet until the publication date (the day on which it is uploaded onto our web site).

Before your manuscript is typeset, we will edit the text to ensure it is intelligible to our wide readership and conforms to house style. We look particularly carefully at the titles of all papers to ensure that they are relatively brief and understandable.

Once your manuscript is typeset and you have completed the appropriate grant of rights, you will receive a link to your electronic proof via email with a request to make any corrections within 48 hours. If, when you receive your proof, you cannot meet this deadline, please inform us at rjsproduction@springernature.com immediately.

If you have queries at any point during the production process then please contact the production team at rjsproduction@springernature.com. Once your paper has been scheduled for online publication, the Nature press office will be in touch to confirm the details.

Content is published online weekly on Mondays and Thursdays, and the embargo is set at 16:00 London time (GMT)/11:00 am US Eastern time (EST) on the day of publication. If you need to know the exact publication date or when the news embargo will be lifted, please contact our press office after you have submitted your proof corrections. Now is the time to inform your Public Relations or Press Office about your paper, as they might be interested in promoting its publication. This will allow them time to prepare an accurate and satisfactory press release. Include your manuscript tracking number NATCOMPUTSCI-22-0681B and the name of the journal, which they will need when they contact our office.

About one week before your paper is published online, we shall be distributing a press release to news organizations worldwide, which may include details of your work. We are happy for your institution or funding agency to prepare its own press release, but it must mention the embargo date and Nature Computational Science. Our Press Office will contact you closer to the time of publication, but if you or your Press Office have any inquiries in the meantime, please contact press@nature.com.

We welcome the submission of potential cover material (including a short caption of around 40 words) related to your manuscript; suggestions should be sent to Nature Computational Science as electronic files (the image should be 300 dpi at 210 x 297 mm in either TIFF or JPEG format). We also welcome suggestions for the Hero Image, which appears at the top of our [home page](http://www.nature.com/natcomputsci); these should be 72 dpi at 1400 x 400 pixels in JPEG format. Please note that such pictures should be selected more for their aesthetic appeal than for their scientific content, and that colour images work better than black and white or

grayscale images. Please do not try to design a cover with the Nature Computational Science logo etc., and please do not submit composites of images related to your work. I am sure you will understand that we cannot make any promise as to whether any of your suggestions might be selected for the cover of the journal.

Best regards,

Ananya Rastogi, PhD
Associate Editor
Nature Computational Science

P.S. Click on the following link if you would like to recommend Nature Computational Science to your librarian: https://www.springernature.com/gp/librarians/recommend-to-your-library

** Visit the Springer Nature Editorial and Publishing website at www.springernature.com/editorial-and-publishing-jobs for more information about our career opportunities. If you have any questions please click here. **